# Fully Spiking Neural Networks for Unified Frame-Event Object Tracking

**Jingjun Yang**[*]    **Liangwei Fan**[*]  **Jinpu Zhang**[†] **Xiangkai Lian**   **Hui Shen**[†]   **Dewen Hu**

College of Intelligence Science and Technology

National University of Defense Technology

{yjingjun, fanliangwei, zhangjinpu, lianxiangkai, shenhui, dwhu}@nudt.edu.cn

Codes and models: https://github.com/Noctis-A/SpikeFET

## Abstract

The integration of image and event streams offers a promising approach for achieving robust visual object tracking in complex environments. However, current fusion methods achieve high performance at the cost of significant computational overhead and struggle to efficiently extract the sparse, asynchronous information from event streams, failing to leverage the energy-efficient advantages of event-driven spiking paradigms. To address this challenge, we propose the first fully Spiking Frame-Event Tracking framework called SpikeFET. This network achieves synergistic integration of convolutional local feature extraction and Transformer-based global modeling within the spiking paradigm, effectively fusing frame and event data. To overcome the degradation of translation invariance caused by convolutional padding, we introduce a Random Patchwork Module (RPM) that eliminates positional bias through randomized spatial reorganization and learnable type encoding while preserving residual structures. Furthermore, we propose a Spatial-Temporal Regularization (STR) strategy that overcomes similarity metric degradation from asymmetric features by enforcing spatio-temporal consistency among temporal template features in latent space. Extensive experiments across multiple benchmarks demonstrate that the proposed framework achieves superior tracking accuracy over existing methods while significantly reducing power consumption, attaining an optimal balance between performance and efficiency.

## 1 Introduction

Visual target tracking has important application value in areas such as automatic driving and intelligent surveillance. Although traditional frame-based tracking methods [1, 2, 3, 4] demonstrate satisfactory performance in conventional scenarios, robust tracking under complex environments involving low-light conditions, over-exposure, and high-speed motion remains a challenge. Event cameras offer new possibilities for tracking tasks due to their high dynamic range, microsecond response, and low power consumption [5], but their sparse and asynchronous characteristics makes it difficult to capture target appearance features. Some researchers [6, 7, 8] have attempted to improve the performance and robustness of tracking by integrating motion information from frames with appearance information from events. However, most Artificial Neural Networks (ANNs)-based methods rely on dense texture information for tracking and struggle to effectively model sparse event data. In contrast, Spiking Neural Networks (SNNs) demonstrate exceptional compatibility and ultra-low energy consumption advantages in event stream processing through their bio-inspired spatiotemporal dynamics and event-driven sparse computational paradigm [9, 10]. Recently, an

---

[*]These authors contributed equally to this work.

[†]Correspondence

39th Conference on Neural Information Processing Systems (NeurIPS 2025).

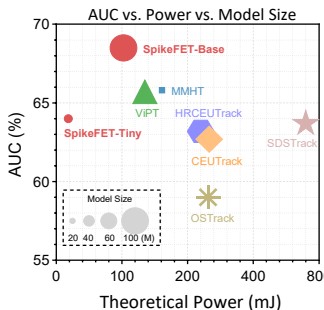

Figure 1: SpikeFET versus other tracking methods on COESOT.

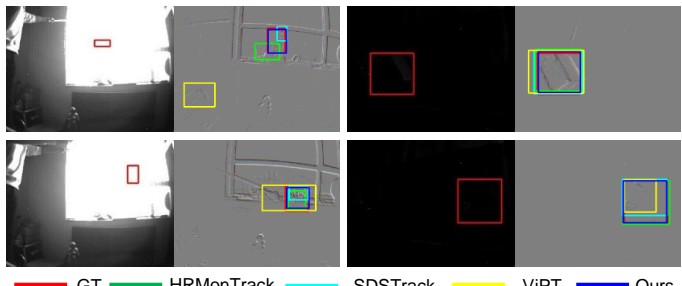

Figure 2: Visualization results of SpikeFET-Base in low light and overexposure scenarios.

E-SpikeFormer [11] based on Metaformer [12] demonstrated that SNNs can achieve comparable performance to classical ANN-based vision transformer models in classification tasks. However, the application of SNNs in object tracking remains underdeveloped, facing critical bottlenecks in architecture-modality co-optimization. Current research predominantly focuses on ANN-SNN hybrid architectures for unimodal tracking [13, 14]. While [15] extends this framework to multimodal tracking, it suffers from compromised energy efficiency. Notably, SDTrack [16] implements a fully spike-driven network but remains constrained to unimodal event-stream inputs, with interpolated noise in feature alignment causing error accumulation and performance degradation.

To address these challenges, we develop a novel fully **Spiking** **F**rame-**E**vent **T**racking framework, named **SpikeFET**. We develop a hierarchical tracking architecture with dual-branch feature extraction, single-branch fusion and dual-branch prediction modules. This design effectively integrates local feature representation of convolutional networks with global modeling capabilities of Transformers, enhancing cross-modal fusion performance. In order to address the adverse impact of positional bias introduced by padding [17, 18] in the convolution module on tracking accuracy, we propose a simple yet effective Randomized Patchwork Module (RPM). This module innovatively reorganizes target spatial distributions by randomly composing initial templates, online updated templates, and search frames into fused rectangular patches, while introducing learnable type encoding to explicitly identify the logical positions of each image patch. Without removing padding in residual units, RPM effectively mitigates padding-induced degradation of translation invariance, significantly enhancing model robustness against target positional variations. In addition, to further address the asymmetric boundary features of adjacent temporal template frames caused by RPM. we develop a Spatial-Temporal Regularization (STR) strategy that enhances both the temporal consistency between dual template target features in the latent space and the spatial consistency of boundary features, thereby improving the robustness of similarity measurement. In particular, SpikeFET-Tiny achieves a 2.7% higher AUC score on the FE108 [8] dataset compared to the state-of-the-art SDSTrack [19], while demonstrating 39× lower power consumption.

In summary, the main contributions of this paper can be summarized as follows:

- We propose a unified frame-event spiking tracker termed SpikeFET. To the best of our knowledge, it is the first work that employs a fully spiking neural network for unified frame-event object tracking.

- We propose a simple yet effective RPM that mitigates positional bias introduced by convolutional padding through randomized spatial reorganization, thereby significantly enhancing the model's robustness to target position variations.

- We develop a STR strategy that overcomes similarity degradation from asymmetric features by enforcing spatio-temporal consistency among temporal template features in latent space, enhancing the tracking accuracy and stability of the model.

- Extensive experimental validation on multiple frame-event tracking benchmarks fully validates the effectiveness of our proposed SpikeFET, and achieves an optimal balance between computational power consumption and performance and parameters, as shown in Fig. 1.

## 2 Related Work

**Visual Object Tracking.** In recent years, deep learning has driven remarkable advancements in visual target tracking. Current mainstream approaches fall into three categories: First, CNN-based trackers like [1, 17] utilize convolutional backbones for feature extraction and employ correlation mechanisms for target localization. However, their limited receptive fields restrict long-range dependency modeling, leading to reduced robustness in scenarios with rapid motion and occlusions. Second, hybrid CNN-Transformer frameworks exemplified by TransT [2] and TMT [20] address this limitation by incorporating self-attention mechanisms into convolutional feature encoding, significantly improving spatio-temporal context modeling through complementary architecture design. Third, pure Transformer-based trackers [3, 21] demonstrate superior global relationship capture capabilities by unifying feature extraction and interaction within an attention-driven framework, showing promising potential for comprehensive context understanding.

**Frame-Event based Object Tracking** With the notable strengths of event data in low-light, fast-motion, and other complex scenes, frame-event fusion methods for single-object tracking have been gradually gaining attention in recent years. Some approaches [6, 7, 13, 22, 23] are based on cross-modal fusion frameworks, like CEUTrack [7], simplify traditional two-branch architectures with a unified Transformer for synchronized multimodal feature extraction. TENet [23] improves event feature representation with lightweight multi-scale pooling and cross-modal mutual guidance for enhanced complementarity. Meanwhile, other methods [24, 25] introduce prompt learning strategies, such as ViPT [25], which refines event features using pre-trained image models and attention mechanisms. In this paper, we propose a dual-single-dual framework for frame-event fusion, featuring independent feature extraction branches for each modality in the early stage and decoupled tracking heads after fusion to achieve efficient inter-modal complementarity and precise localization.

**Spiking Neural Networks for Object Tracking** SNNs show superior performance and efficiency in vision tasks [11, 26, 27, 28] through event-driven, low-power operation, excelling in time-sensitive domains like dynamic object tracking with bio-inspired asynchronous computation. Early frame-based SNN trackers [29, 30, 31] integrated Siamese networks with temporal encoding using multi-step methods, and then compressed SNNs into single step models using knowledge distillation. For event-driven tracking [13, 14, 32, 33], STNet [13], and SNNTrack [14] enhanced spatiotemporal feature extraction but faced challenges in scenarios with insufficient texture information. SDTrack [16] is the first transformer-based spike-driven tracker, but also lacks robustness to spatial offsets. Frame-event fusion models MMHT [15] utilized ANN-SNN hybrid architectures for spatiotemporal integration but did not fully exploit the energy-efficient advantages of SNNs. To overcome these limitations, we present SpikeFET, the first fully spiking frame-event tracker. By adopting a fully SNN-based multimodal fusion framework, SpikeFET fully unleashes the spatiotemporal modeling capabilities of spiking networks while preserving their energy-efficient computational properties.

## 3 Methodology

As shown in Fig. 3, the overall framework of SpikeFET adopts a dual-single-dual structure based on SNN transformer. First, the image frames and event temporal frames are randomly combined using the RPM to mitigate the padding effects during feature extraction by convolutional layers. Subsequently, the combined multi-modal images are fed into a feature extraction network, where ConvFormer Spike Blocks are employed to comprehensively extract features. Positional encoding, modality encoding, and type encoding are then incorporated into the feature embeddings, followed by direction-consistent concatenation of the two modal images. The TransFormer Spike Block jointly processes and correlates these embeddings. Meanwhile, STR is introduced for dual template frames to maintain spatiotemporal consistency. Finally, the resulting feature embeddings are delivered to an SNN-based center-tracking head for prediction, with a constraint to ensure consistency between the response maps generated by both modalities.

### 3.1 Spiking Frame-Event Tracking Network

In order to enhance the cross-modal feature fusion capability, we design a framework that integrates a dual branch feature extraction network and a dual modal response graph fusion mechanism. This framework consists of three core modules: a dual branch modal specific feature extraction network, a

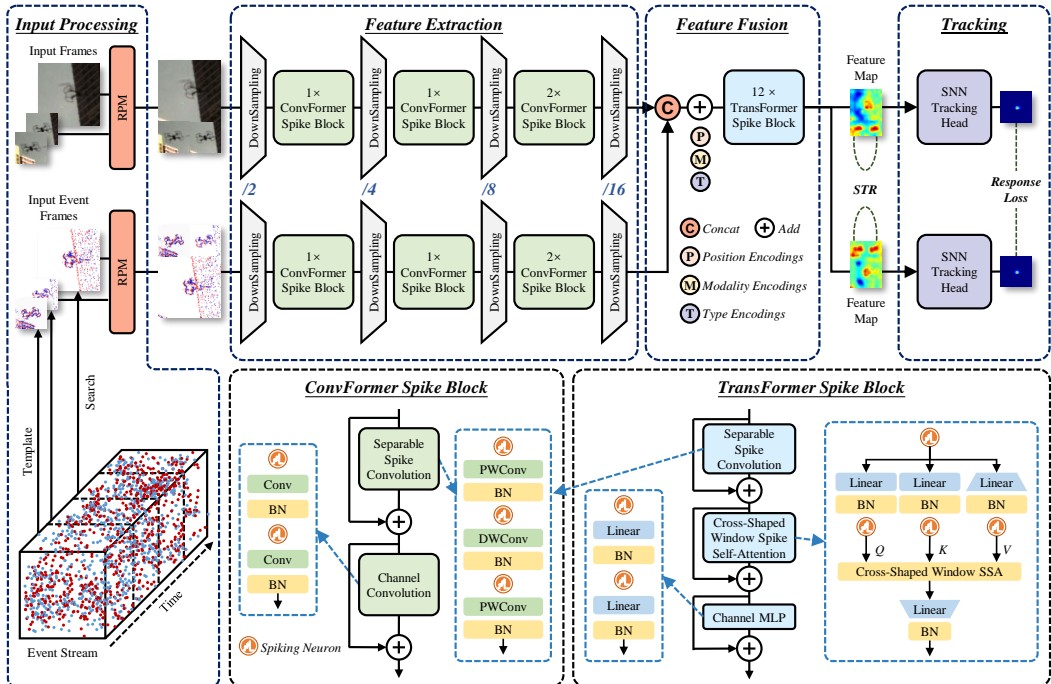

Figure 3: The overview of SpikeFET. Our SpikeFET consists of Input Processing, Feature Extraction, Feature Fusion and Tracking. Where the Event Stream is simply transformed into event temporal frames using [34]. Feature Extraction consists of DownSampling and ConvFormer Spike Block cascades. Feature Fusion consists of TransFormer Spike Block cascades. Tracking uses SNN Tracking Head, and Spiking Neuron use the Spike Firing Approximation (SFA) [11]

cross-modal fusion network, and a dual head decoupling tracking prediction with similarity fusion, as detailed in Fig. 3.

**Modality-Specific Feature Extraction Network** The proposed dual-branch modality-specific feature extraction network consists of a image branch and an event branch operating in parallel, where each branch comprises multiple stacked ConvFormer Spike blocks. Specifically, the ConvFormer Spike Block can be formulated as:

$$\mathbf{U}' = \mathbf{U} + \text{SepSpikeConv}(\mathbf{U}) \tag{1}$$

$$\mathbf{U}'' = \mathbf{U}' + \text{ChannelConv}(\mathbf{U}') \tag{2}$$

$$\text{SepSpikeConv}(\mathbf{U}) = \text{Conv}_{\text{pw2}}(\text{SN}(\text{Conv}_{\text{dw}}(\text{SN}(\text{Conv}_{\text{pw1}}(\text{SN}(\mathbf{U})))))) \tag{3}$$

$$\text{ChannelConv}(\mathbf{U}') = \text{Conv}(\text{SN}(\text{Conv}(\text{SN}(\mathbf{U}')))) \tag{4}$$

where SepSpikeConv($\cdot$) denotes the token mixer, ChannelConv($\cdot$) represents the channel mixer, Convpw1($\cdot$) and Convpw2($\cdot$) are pointwise convolutions, Convdw($\cdot$) refers to depthwise convolution [35], and Conv($\cdot$) indicates standard convolution. SN($\cdot$) stands for the spiking neuron layer. Note that BN layers are omitted here for notation simplicity.

**Cross-Modal Fusion Network** Before feeding the dual-modality feature maps into the fusion network, we introduce three standard learnable encodings: positional encoding $E_p$, modality encoding $E_m$, and type encoding $E_t$ (see Sec.3.2). The dual-modality feature maps are then summed with their corresponding three encodings:

$$\mathbf{U}' = \mathbf{U} + E_p + E_m + E_t \tag{5}$$

The image feature map $U_i$ and the event feature map $U_e$ are then concatenated as $U = [U_i; U_e]$. The generated feature map U serves as the input to the fusion network for cross-modal interactive learning.

The cross-modal fusion network consists of Transformer Spike Blocks, which can be formulated as:

$$\mathbf{U}' = \mathbf{U} + \text{SepSpikeConv}(\mathbf{U}) \tag{6}$$

$$\mathbf{U}'' = \mathbf{U}' + \text{CSWin-SSA}(\mathbf{U}') \tag{7}$$

$$\mathbf{U}''' = \mathbf{U}'' + \text{ChannelMLP}(\mathbf{U}'') \tag{8}$$

where CSWin-SSA$(\cdot)$ denotes the Cross-Shaped Windows [36] Spiking Self-Attention module. Specifically, the CSWin-SSA module can be expressed as:

$$\mathbf{Q_S} = \text{SN}(\text{Linear}(\mathbf{U})), \;\; \mathbf{K_S} = \text{SN}(\text{Linear}(\mathbf{U})), \;\; \mathbf{V_S} = \text{SN}(\text{Linear}_\gamma(\mathbf{U})) \tag{9}$$

$$\mathbf{U}' = \text{Linear}_{\frac{1}{\gamma}}(\text{CSWinSSA}(\mathbf{Q_S}, \mathbf{K_S}, \mathbf{V_S})) \tag{10}$$

$$\text{SSA}(\mathbf{Q_S}, \mathbf{K_S}, \mathbf{V_S}) = \text{SN}(\mathbf{Q_S}\mathbf{K_S}^\top \mathbf{V_S} * \text{scale}) \tag{11}$$

where Linear$(\cdot)$ denotes a linear transformation, $\gamma$ is the channel expansion factor [37], CSWinSSA$(\cdot)$ represents the CSWinSSA operator, and SSA$(\cdot)$ is the spiking self-attention operator within the CSWinSSA operator. To address the challenge of large values generated by matrix multiplication, a scaling factor (scale) is introduced to regulate the magnitude of the results. Further details of the CSWinSSA operator are provided in the Appendix E.

**Decoupled Tracking Prediction With Similarity Fusion** In the design of the tracking head, we maintain structural continuity with the dual-modal feature extractor. Specifically, we construct the Decoupled Spike Tracking Heads similar to SDTrack [16], utilizing parallel dual branches to process heterogeneous modal features.

In object tracking tasks, the response maps output by the network play a critical role in target localization. To achieve sufficient fusion of image and event modalities, we propose enhancing feature alignment by constraining the similarity between the response maps of the two modalities, implemented via a weighted focal loss function $\mathcal{L}_{\text{GWF}}$ [38] (see the Appendix C.1 for more details). The final loss function is expressed as: $\mathcal{L}_{\text{Res}} = \mathcal{L}_{\text{GWF}}(\mathbf{R}_F/\tau, \mathbf{R}_E/\tau)$, where $\mathbf{R}_F$ and $\mathbf{R}_E$ are the response maps of two modes, $\tau$ is the temperature coefficient, and the empirical setting is 2.

### 3.2 Randomized Patchwork Module

The widely used padding in residual networks [17, 18] significantly degrades single-object tracking performance by breaking translation invariance, while traditional methods of removing padding [17, 18] would disrupt the network structure. To address this critical issue, while preserving the advantages of the original network core architecture, we have innovatively proposed a simple yet effective Randomized Patchwork Module (RPM). Specifically, prior to modality-specific feature extraction, Two adjacent temporal template images $Z_1, Z_2 \in \mathbb{R}^{T \times C \times H_z \times W_z}$ and the search image $X \in \mathbb{R}^{T \times C \times H_x \times W_x}$ are fed into the RPM. As shown in Fig. 4, RPM randomly employs either horizontal concatenation $U_h \in \mathbb{R}^{T \times C \times H_z \times (W_z + W_x)}$ or vertical concatenation $U_v \in \mathbb{R}^{T \times C \times (H_z + H_x) \times W_z}$ to generate rectangular fused inputs. This design randomly distributes the search image across four orientations (top, bottom, left, right) of the concatenated region, creating diversified input patterns. Meanwhile, we enforce consistent concatenation direction (horizontal or vertical) between image frames and event streams to ensure spatial consistency across modalities.

Additionally, as shown in Fig. 3, the learnable type encoding $E_t$ explicitly identifies the logical positions of image patches through additive integration with visual features, eliminating spatial ambiguities caused by random patch permutation while suppressing noise. Through visual analysis of the heatmap distribution in output images (Fig. 6), the effectiveness of the module is clearly demonstrated.

### 3.3 Spatial-Temporal Regularization

Although RPM dynamically reorganizes the spatial distribution of input data to mitigate target localization bias, the boundary regions of adjacent temporal template frames (e.g., $Z_1$ and $Z_2$) still manifest asymmetric feature sources, such as distinct features from other images or padding regions induced by convolutional networks. As illustrated in Fig. 5, the left boundary of $Z_1$ corresponds to padding regions, while the left boundary of $Z_2$ coincides with the right region of $Z_1$, while their bottom boundaries respectively align with different regions of the search frame X. This spatial distribution

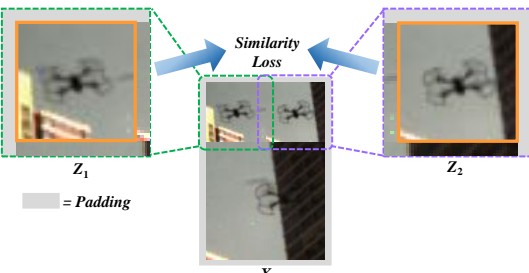

Figure 4: Detailed design of proposed RPM. Randomly combine template frames $Z_1$ and $Z_2$ with search frame X.

Figure 5: Illusion of the proposed STR. Applying similarity loss regularization to two temporally adjacent template frames.

inconsistency inherently disrupts the strict translation invariance of convolutional networks, leading to feature representation deviations at identical spatial positions across template frames $Z_1$ and $Z_2$ and consequently causing significant degradation in similarity measurement. Notably, we also found that constraining the similarity of latent feature space between adjacent template frames in the temporal domain can establish a more robust similarity metric space.

Building upon the aforementioned insights, we propose the Spatial-Temporal Regularization (STR) strategy to constrain the similarity measurement of dual-template frame features. The core idea behind this strategy is to leverage two temporally adjacent template frame features with spatial-temporal consistency for relation modeling with the search frame features, thereby enhancing model robustness. Specifically, after the cross-modal fusion network completes multi-modal information integration, we construct a similarity constraint on the fused features output by temporally adjacent template frames. By applying mean squared error (MSE) to regularize the feature discrepancies between dual templates, we establish a similarity supervision signal with spatial-temporal consistent characteristics. This constraint can be formulated as the following loss function:

$$\mathcal{L}_{\text{sim}} = \frac{1}{hw} \sum_{i=1}^{h} \sum_{j=1}^{w} \left( F_1^{(i,j)} - F_2^{(i,j)} \right)^2 \tag{12}$$

where $F_1 \in \mathbb{R}^{h \times w}$ and $F_2 \in \mathbb{R}^{h \times w}$ denote the output feature maps of two temporally adjacent template frames after passing through the feature fusion network, where $h$ and $w$ represent the spatial dimensions of the feature maps. The loss function guides the network to extract more generalizable feature representations from spatiotemporally continuous video data by minimizing the discrepancies between spatially corresponding points in the dual-template features.

### 3.4 Training and Inference

For training, we adopt the same training procedure as OSTrack [3], employing weighted focal loss [38] for classification and a combination of L1 loss and generalized IoU loss [39] for regression. For cross-modal fusion and the STR strategy, we employ the similarity-based loss functions $\mathcal{L}_{\text{Res}}$ and $\mathcal{L}_{\text{Sim}}$ as previously described. Finally, the overall loss function is formulated as:

$$\mathcal{L} = \mathcal{L}_{\text{cls}}^F + \lambda_{\text{iou}}\mathcal{L}_{\text{iou}}^F + \lambda_{\mathcal{L}1}\mathcal{L}_{\text{L1}}^F + \mathcal{L}_{\text{cls}}^E + \lambda_{\text{iou}}\mathcal{L}_{\text{iou}}^E + \lambda_{\mathcal{L}1}\mathcal{L}_{\text{L1}}^E \tag{13}$$

$$\mathcal{L}_{\text{track}} = \mathcal{L} + \alpha\mathcal{L}_{\text{res}} + \beta\mathcal{L}_{\text{sim}} \tag{14}$$

where $\lambda_{\text{iou}} = 2$ and $\lambda_{\text{L1}} = 5$ are regularization parameters as in [4]. $\alpha = 1$ and $\beta = 0.5$ are hyperparameters used to balance the contributions of different loss functions. Ablation studies on hyperparameters are shown in Appendix F.

For inference, we perform a weighted summation and fusion of the image frame and event temporal frame response maps output by the dual-head decoupled tracking head to produce robust target localization results. This is expressed as:

$$\mathcal{R} = \lambda\mathcal{R}_{\text{F}} + (1 - \lambda)\mathcal{R}_{\text{E}} \tag{15}$$

where $\lambda = 0.5$ is a hyperparameter to balance the responses of different modalities. It should be noted that we do not employ a dynamic template update strategy. Instead, we consistently replicated

the first frame to use two template frames (both derived from the first frame) and one search frame for inference. At the same time, we follow common practice by applying a Hanning window penalty to leverage positional priors in tracking [2, 3].

# 4 Experiments

## 4.1 Implementation details

Our proposed SpikeFET was implemented with PyTorch 1.12 in Python 3.8 and trained on two NVIDIA RTX 4090 GPUs. The network input consists of a triplet image group comprising one search image and two distinct template images. When constructing training samples from randomly sampled video sequences, we expanded the search regions and template regions to 4× and 2× their original bounding box sizes, respectively, followed by resizing them to 256×256 and 128×128. Common data augmentation techniques such as horizontal flipping and brightness jittering were applied to the image sets. The model was trained using the AdamW optimizer. The optimizer has a cosine annealing scheduling over 50 epochs, where each epoch contained 60,000 image triplets. Please refer to the Appendix D for more details.

Similar to current mainstream multi-modal object tracking frameworks (such as VIPT [25], SD-STrack [19], TENet [23]), which commonly adopt OSTrack [3] frame-datasets pre-trained models for parameter fine-tuning or knowledge distillation, we also follow their standard configuration process. Specifically, we adapt our SpikeFET into a single-modal SpikeET model, pre-trained on frame-datasets such as COCO [40], LaSOT [41], TrackingNet [42], and GOT-10K [43] to fine-tune our SpikeFET network. For the single-modal SpikeET model, we remove one feature extraction branch and the spiking tracking head branch from SpikeFET, along with eliminating modality encoding. To balance speed and accuracy, we propose three variants with distinct architectures and parameters: SpikeFET-Base (Fig. 3) with larger parameter capacity, and SpikeFET-Tiny and SpikeET optimized for computational efficiency. Among these, only SpikeFET-Base undergoes frame-based dataset pre-training, while SpikeFET-Tiny and SpikeET adopt ImageNet-1K [44] to pre-train the backbone, consistent with OSTrack [3]. To quantitatively evaluate the performance of tracking systems, we employ three core metrics: Area Under the Curve (AUC), Precision Rate (PR), and theoretical power consumption. For a detailed description of the theoretical power consumption calculation method, please refer to Appendix C.2.

## 4.2 Datasets

We evaluated our proposed SpikeFET using three large-scale frame-event single-object tracking datasets: FE108 [8], VisEvent [6], and COESOT [7]. All three datasets were captured using the DAVIS346 camera, which features a spatial resolution of 346 × 260, a dynamic range of 120 dB, and a minimum latency of 20 μs.

Specifically, The FE108 dataset [8] contains 108 synchronized frame-event sequences captured in indoor environments, totaling approximately 1.5 hours of data. It covers 21 distinct target objects and is divided into 76 training sequences and 32 testing sequences. The target bounding box annotations were generated using a Vicon motion capture system. The VisEvent [6] dataset originally collected 820 frame-event pairs, divided into 500 training sequences and 320 testing sequences. Following [13], we removed sequences with missing event data or misaligned timestamps, resulting in a refined dataset of 205 training sequences and 172 testing sequences. Multimodal networks and unimodal networks use 320 test sequences and 172 test sequences, respectively. COESOT [7], as a larger-scale benchmark, contains 578K frame-event pairs divided into 827 training sequences and 527 testing sequences. Collected across diverse indoor and outdoor scenarios, it covers 90 target categories annotated with 17 challenging visual attributes. All target bounding boxes were manually annotated.

For fair comparison, we trained SpikeFET using the full training set of VisEvent and evaluated it on the test set containing 320 sequences, while SpikeET was trained on a pruned version of the VisEvent training set and tested on a subset of 172 sequences. Note that on SpikeFET, we only use raw event frames from the dataset on VisEvent, not event temporal frames.

## 4.3 Main Results

We evaluated the performance of our proposed SpikeFET-Tiny and SpikeFET-Base on several benchmarks, including FE108 [8], VisEvent [6], and COESOT [7]. Meanwhile, for comparison with event-based trackers, we constructed a baseline model named SpikeET that using only event streams as input.

Table 1: Comparison with state-of-the-art trackers on FE108 [8], VisEvent [6], and COESOT [7]. We train and measure energy consumption on the VisEvent dataset. The best three results are shown in **red**, **blue** and **green** fonts. ∗ indicates that frame-based trackers are extended to frame-event-based trackers through early fusion. † indicates models pre-trained on image-frame tracking datasets.

| Methods | Architecture | Param (M) | Power (mJ) | FE108 [8] | | VisEvent [6] | | COESOT [7] | |
|---|---|---|---|---|---|---|---|---|---|
| | | | | AUC(%) | PR(%) | AUC(%) | PR(%) | AUC(%) | PR(%) |
| ANN | DiMP50∗ [45] | - | - | 57.1 | 85.1 | 47.8 | 67.0 | 58.9 | 67.1 |
| | PrDiMP50∗ [46] | - | - | 59.0 | 87.7 | 47.6 | 65.3 | 57.9 | 69.6 |
| | SiamRCNN∗ [47] | - | - | - | - | 49.9 | 65.9 | 60.9 | 71.0 |
| | TrDiMP50∗ [20] | - | - | 60.3 | 91.2 | - | - | 60.1 | 72.2 |
| | TransT50∗ [2] | - | - | 63.9 | 93.0 | 47.4 | 65.0 | 60.5 | 72.4 |
| | ToMP101∗ [48] | - | - | 61.8 | 91.1 | - | - | 59.9 | 67.2 |
| | FENet [8] | - | 262.2 | 63.1 | 91.8 | - | - | - | - |
| | OSTrack [3] | 92.52 | 262.29 | - | - | 53.4 | 69.5 | 59.0 | 70.7 |
| | CEUTrack [7] | 97.82 | 265.60 | 55.6 | 84.5 | 53.1 | 69.1 | 62.7 | 76.0 |
| | HRCEUTrack [22] | 97.82 | 239.79 | - | - | - | - | 63.2 | 71.9 |
| | HRMonTrack [22] | 100.20 | - | **68.5** | **96.2** | - | - | - | - |
| | ViPT† [25] | **93.36** | **134.93** | 65.2 | 92.1 | **59.2** | **75.8** | **65.7** | **73.9** |
| | SDSTrack† [19] | 107.80 | 719.10 | **65.8** | 92.6 | **59.7** | **76.7** | 63.7 | 71.7 |
| ANN-SNN | MMHT [15] | **22.80** | 160.93 | 63.0 | 93.6 | 55.1 | 73.3 | **65.8** | 74.0 |
| SNN | **SpikeFET-Tiny** | **29.33** | **18.36** | **68.5** | **96.5** | 56.8 | 73.5 | 64.0 | **77.9** |
| | **SpikeFET-Base**† | 105.48 | **102.61** | **68.7** | **97.0** | **59.0** | **75.3** | **68.5** | **81.7** |

**Frame-Event Tracking** As shown in Tab. 1, compared to several state-of-the-art trackers, our SpikeFET demonstrates significant advantages across most datasets. On the FE108 [8] dataset, SpikeFET-Tiny achieves an AUC score of 68.5%, matching the previous state-of-the-art HRMon-Track [22] while surpassing it by 0.3% in PR score and outperforming all other trackers by a substantial margin. Compared to the Tiny variant, the Base model further improves performance metrics, but there is a tendency towards overfitting. Notably, although the model size of the base model is comparable to the previous Base models, its theoretical energy consumption is significantly lower than equivalently scaled models. Additionally Compared with the latest SDSTrack [19] on ANN, our Tiny has improved its AUC score by 2.7% on the FE108 [8] dataset, with parameters only one-third of SDSTrack [19], while reducing power consumption by 39 times.

On the more complex COESOT [7] dataset, our SpikeFET-Base model achieves 2.7% and 7.7% improvements in AUC and PR scores respectively compared to the state-of-the-art MMHT [15], while the Tiny model also surpass the latest SDSTrack [19] by 0.3% in AUC. This demonstrates that our method exhibits stronger capabilities in handling large-scale complex scenarios.

On the VisEvent [6] dataset, our Tiny model outperforms the majority of trackers. However, for the Base model, our SpikeFET performs comparatively worse than ViPT [25] and SDSTrack [19]. We attribute this primarily to the fact that SpikeFET is particularly well-suited for handling datasets with challenging scenarios and large-scale datasets, such as FE108 [8] and COESOT [7]. In such scenarios, the inherent advantages of event cameras become evident, allowing SpikeFET to demonstrate exceptional performance. In contrast, ViPT [25] and SDSTrack [19] excel in less challenging scenarios, such as VisEvent [6].

At the same time, as shown in Tab. 1, it can be observed that SpikeFET-Base achieved increases of 0.2%, 2.2%, and 4.5% in AUC metrics compared to SpikeFET-Tiny on the FE108 [8], VisEvent [6], and COESOT [7] datasets (arranged in ascending order of data volume), with the performance gap progressively widening. Tab. 2 reveals that on FE108 [8]—a dataset featuring numerous challenging scenarios (e.g., overexposure, low illumination)—SpikeFET's performance is fully leveraged: its AUC is only 4.1% lower than SpikeFET-Tiny, while outperforming all event-based tracking algorithms. In contrast, it underperforms by 17.4% on VisEvent [6]. This further illustrates the superiority of SpikeFET in challenging scenarios and complex datasets.

Table 2: Comparison with state-of-the-art trackers on two event-based tracking benchmarks. The works in the table directly adopt the results of the SDTrack [16] report.

| Methods | Architecture | Param (M) | Power (mJ) | FE108 [8] | | VisEvent [6] | |
|---|---|---|---|---|---|---|---|
| | | | | AUC(%) | PR(%) | AUC(%) | PR(%) |
| ANN | DiMP50 [45] | - | 256.37 | - | - | 31.5 | 44.2 |
| | PrDiMP50 [46] | - | 258.37 | - | - | 32.2 | 46.9 |
| | ATOM [49] | - | 30.199 | - | - | 28.6 | 47.4 |
| | SiamRPN [50] | - | 203.88 | - | - | 24.7 | 38.4 |
| | STARK [4] | 28.23 | 58.88 | 57.4 | 89.2 | 34.1 | 46.8 |
| | SimTrack [51] | 88.64 | 93.84 | 56.7 | 88.3 | 34.6 | 47.6 |
| | OSTrack [3] | 92.52 | 98.90 | 54.6 | 87.1 | 32.7 | 46.4 |
| | ARTrack [52] | 202.56 | 174.80 | 56.6 | 88.5 | 33.0 | 43.8 |
| | SeqTrack [53] | 90.60 | 302.68 | 53.5 | 85.5 | 28.6 | 43.3 |
| | HiT [54] | 42.22 | **19.78** | 55.9 | 88.5 | 34.6 | 47.6 |
| | GRM [55] | 99.83 | 142.14 | 56.8 | 89.3 | 33.4 | 47.7 |
| | HIPTrack [56] | 120.41 | 307.74 | 50.8 | 81.0 | 32.1 | 45.2 |
| ANN-SNN | STNet [13] | **20.55** | - | **58.5** | **89.6** | 35.0 | 50.3 |
| | SNNTrack [14] | **31.40** | **8.25** | - | - | **35.4** | **50.4** |
| SNN | SDTrack [16] | 107.26 | 30.52 | **59.9** | **91.5** | **37.4** | **51.5** |
| | **SpikeET** | **22.36** | **8.80** | **64.4** | **94.7** | **39.4** | **54.0** |

These results confirm the energy efficiency characteristics of SNNs and their potential for edge devices. Furthermore, due to the inherent compatibility between SNNs and event data, combined with the high-efficiency design of our SpikeFET network, SNNs demonstrate promising potential to surpass ANNs in frame-event tracking tasks.

**Event-based Tracking**  As shown in Tab. 2, without using fusion networks, our SpikeET with RPM and STR strategy outperforms all event-based trackers on both datasets, achieving state-of-the-art performance. Specifically, on the FE108 [8] dataset, SpikeET with 22.36M parameters and ultra-low energy consumption improves AUC by 4.5% and PR score by 3.2% compared to the previous best tracker SDTrack [16], even surpassing most trackers listed in Tab. 1. On the VisEvent [6] dataset, SpikeET achieves 2.0% higher AUC and 2.5% better PR than SDTrack [16]. Notably, while consuming only 0.55mJ more energy than SNNTrack [14], our method delivers a 4.0% AUC improvement. These results effectively demonstrate the advantages of our proposed RPM and STR modules, while further validating the energy efficiency of SNNs and their potential for edge device deployment.

## 4.4 Ablation Study

**Effectiveness of proposed components**  We analyze the effects of the proposed RPM, $E_t$, and STR. Our baseline model, similar to CEU-Track [7], processes six input images from both frame and event modalities with separate feature extraction. These features are concatenated before entering the fusion network, then separated prior to the prediction head and processed through decoupled tracking heads. As shown in Tab. 3, the baseline equipped with RPM (#3) achieves significant improvements. On FE108 [8] with numerous extreme scenarios, RPM substantially increases AUC by 34.7% and PR by 49.9% over the baseline, while on VisEvent [6], it improves AUC by 10.4% and PR by 11.9%. More specifically, illustrated in Fig. 6, without RPM method, the peak response in the output image fails to precisely indicate the target position, exhibiting a significant deviation. In contrast, when RPM is employed, the target area is accurately

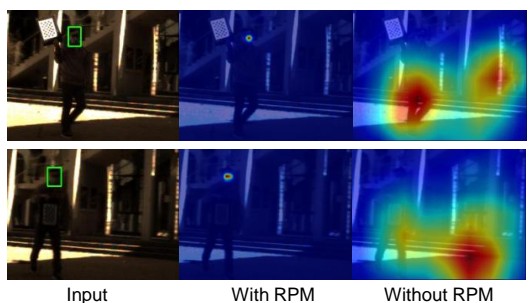

Figure 6: Visualization of position bias learnt in the model w/ and w/o RPM.

highlighted with high response activation. Quantitative analysis confirms that our model achieves enhanced tracking accuracy and robustness through RPM.

Furthermore, we conducted further experiments using one template frame and one search frame per modality, totaling four inputs. Without RPM, the six-input configuration (#baseline) yields only marginally higher performance than the four-input setup (#1) on the FE108 [8] dataset. Similarly, when RPM is employed, the six-input configuration (#3) also shows only a slight performance advantage over the four-input configuration (#2) on the FE108 [8] dataset. This conclusively demonstrates that the primary innovation driving baseline improvement is RPM's effective mitigation of padding-induced degradation in translation invariance. In contrast, utilizing additional data provides minimal performance gains relative to RPM's contribution. Additionally, the six-input setting allows for the further incorporation of STR.

Further additions of $E_t$ (#4) and STR (#5) to #3 demonstrate effective enhancements, particularly with STR improving FE108 [8] AUC by 0.6%. This proves that while retaining the advantage of RPM in dynamically reorganizing spatial distributions, the STR strategy effectively suppresses matching deviations caused by feature spatial misalignment, significantly improving model performance. Finally, integrating both $E_t$ and STR with RPM (#6) yields greater gains: 1.7% AUC and 1.5% PR improvements on FE108 [8], along with 1.3% AUC and 1.1% PR improvements on VisEvent [6].

Table 3: Ablation studies on RPM, $E_t$, and STR (Random Patchwork Module, Type Encodings, Spatio-Temporal Regularization).

| # | Inputs | RPM | $E_t$ | STR | FE108 | | VisEvent | |
|---|---|---|---|---|---|---|---|---|
| | | | | | AUC(%) | PR(%) | AUC(%) | PR(%) |
| **Baseline** | 6 | × | × | × | 32.1 | 45.0 | 45.1 | 60.5 |
| 1 | 4 | × | × | × | 30.8 | 43.9 | 44.3 | 59.2 |
| 2 | 4 | ✓ | × | × | 66.1 | 93.9 | 54.5 | 71.6 |
| 3 | 6 | ✓ | × | × | 66.8 | 94.9 | 55.5 | 72.4 |
| 4 | 6 | ✓ | ✓ | × | 66.8 | 95.0 | 55.7 | 72.3 |
| 5 | 6 | ✓ | × | ✓ | 67.4 | 95.8 | 55.7 | 72.5 |
| 6 | 6 | ✓ | ✓ | ✓ | **68.5** | **96.5** | **56.8** | **73.5** |

Table 4: Ablation studies on the effect of the fusion methods.

| # | Type | Method | FE108 | | VisEvent | |
|---|---|---|---|---|---|---|
| | | | AUC(%) | PR(%) | AUC(%) | PR(%) |
| 1 | **Baseline** | **SpikeFET-Tiny** | **68.5** | **96.5** | **56.8** | **73.5** |
| 2 | Modal | SpikeFT (Frame) | 51.8 | 76.2 | 54.6 | 71.1 |
| 3 | | SpikeET (Event) | 64.4 | 94.7 | 41.0 | 57.1 |
| 4 | | Pre-Concat | 67.5 | 95.9 | 55.6 | 72.3 |
| 5 | Backbone | Concat → Add | 68.0 | 95.4 | 55.2 | 71.9 |
| 6 | | W/o modality Encodings | 68.0 | 96.1 | 55.8 | 72.6 |
| 7 | | W/o Dual-Head | 67.0 | 94.5 | 55.6 | 72.1 |
| 8 | Tracking | W/o Response Loss | 67.1 | 95.5 | 56.1 | 72.9 |
| 9 | | W/o Inference Fusion | 67.9 | 95.6 | 56.1 | 72.8 |

**Effectiveness of the fusion methods**  We evaluate the effectiveness of our fusion approach. As shown in entries #2 and #3 in Tab. 4, our frame-event fusion framework outperforms using either modality alone, demonstrating that our framework can effectively leverage the strengths of both modalities. The results from entry #4 reveal that concatenating the two modal images before feature extraction leads to performance degradation. The underlying reason may be that using independent parameters helps the model distinguish data types, enabling better learning of modality-specific features. In entry #5, we investigate an alternative approach of combining image features with event features via addition. This results in lower performance compared to our default method. Entry #6 demonstrates that modality-specific encoding more effectively differentiates modal features. Tracking experiments (entries #7, #8, and #9) confirm the necessity of our Decoupled Tracking Prediction With Similarity Fusion. Without this design, performance degrades to varying degrees. For more experiments, please refer to Appendix. F

**Visualization results**  We compare the tracking results of SpikeFET-Base with the state-of-the-art tracker shown in Fig. 2. Despite facing challenges such as exposure and low light conditions, our tracker achieves robustness and excellent tracking accuracy. Additional visualization results are provided in the Appendix. A.

## 5   Conclusion

In this article, we propose a full spiking neural network SpikeFET for unified frame-event object tracking. This network effectively integrates frame and event data in the spiking paradigm and achieves collaborative integration of convolutional local feature extraction and Transformer-based global modeling. Moreover, we design the RPM to overcome the degradation of translation invariance caused by convolution filling through random spatial reorganization, while preserving the residual structure. We also propose the STR strategy that overcomes the degradation of similarity metrics caused by asymmetric features by forcing spatiotemporal consistency between temporal template features in the latent space. The effectiveness and power consumption advantages of our SpikeFET have been demonstrated through extensive experiments on multiple benchmarks.

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

# Appendix

## A   More Visualization Results

We present more results of SpikeFET and other methods on the COESOT [7] dataset here.

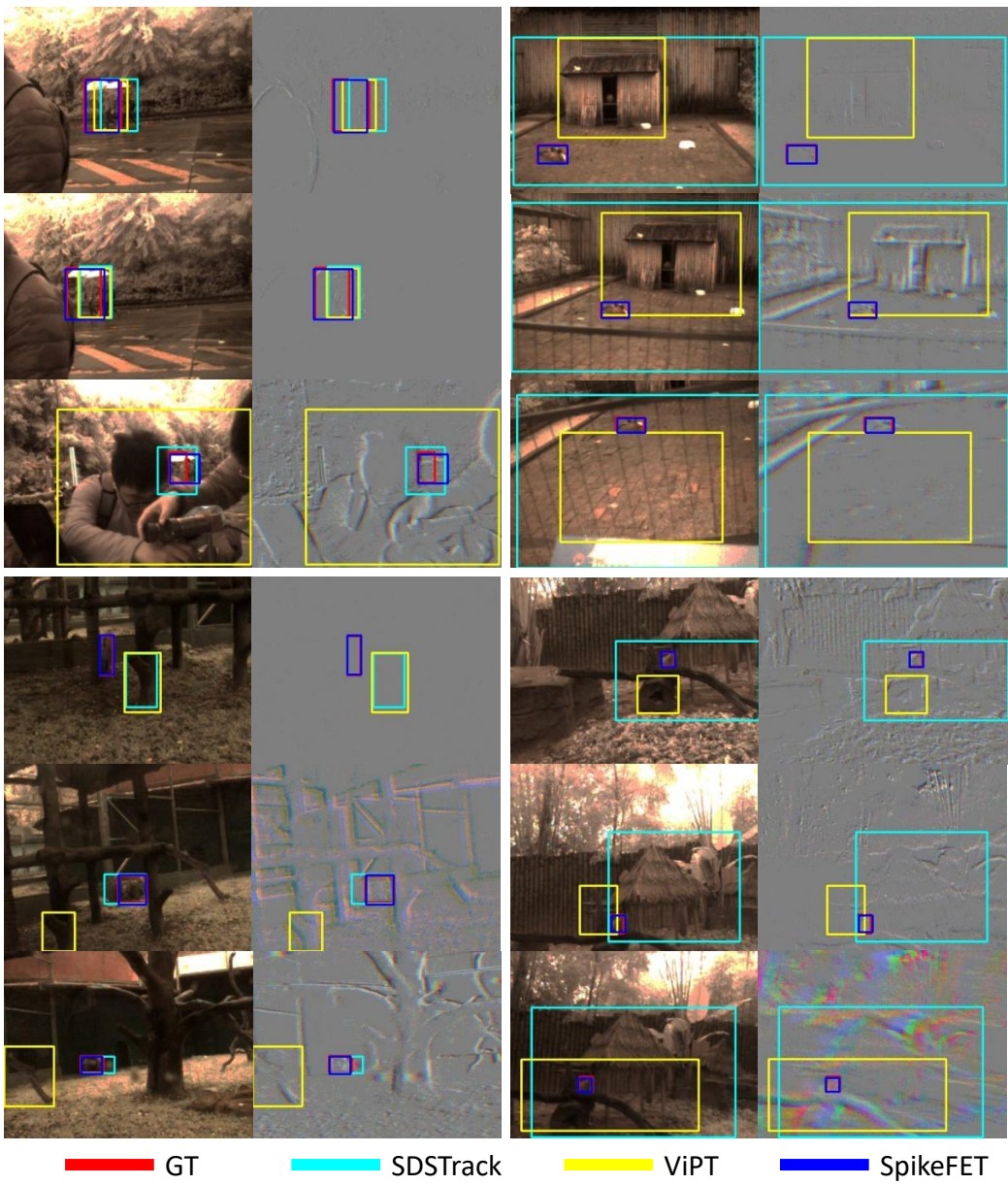

Figure 7: Partial visualization results on the COESOT [7] dataset.

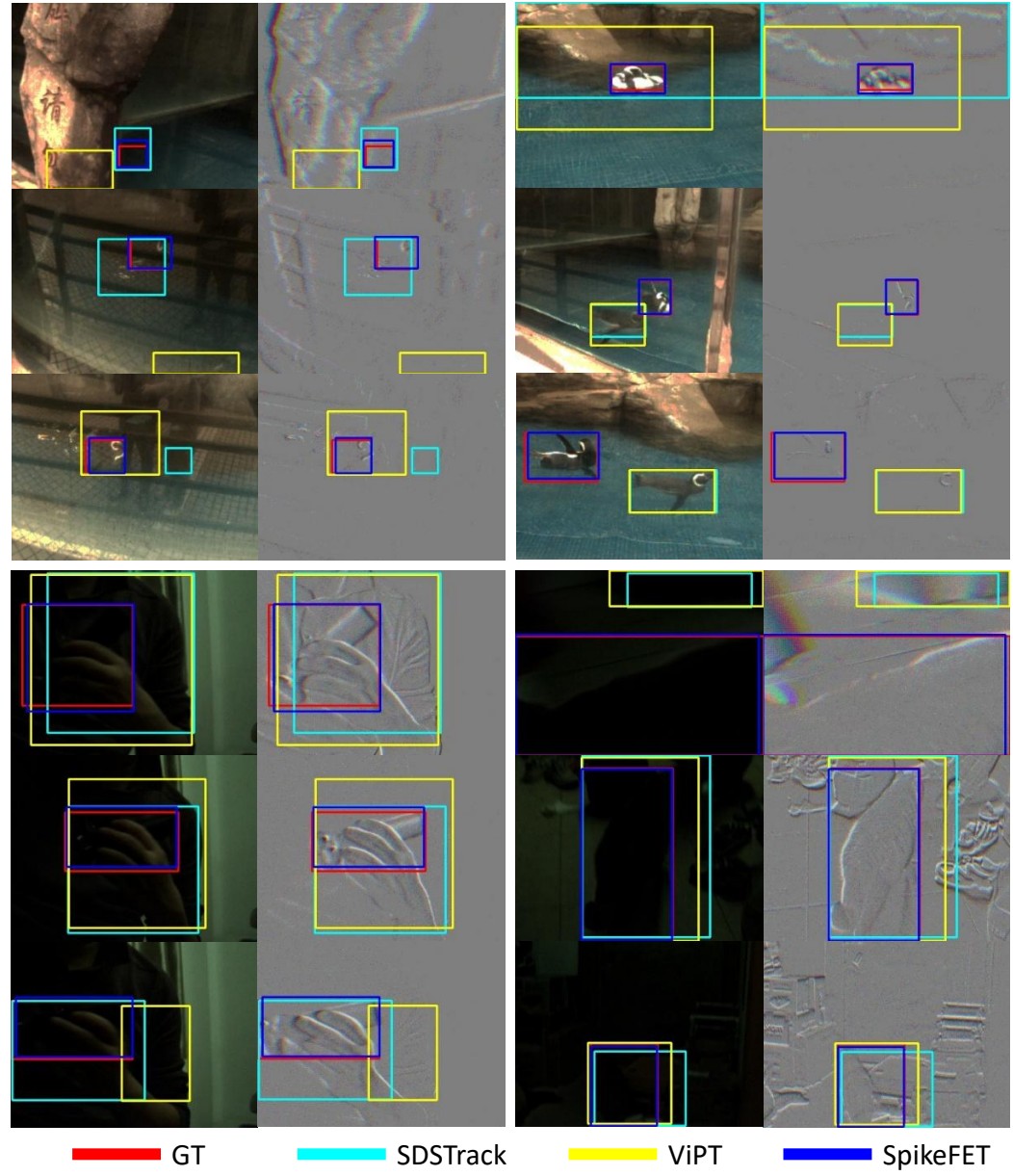

Figure 8: Partial visualization results on the COESOT [7] dataset.

# B  Preliminary

## B.1  Spiking Neuron

SNNs derive biological plausibility from their event-driven operation. Conventional training faces a dilemma: enforcing strict 0,1 spikes induces gradient quantization errors, while mitigating errors via multi-bit spikes [57, 58] or continuous approximations [59] erodes event-driven efficiency. We introduce the Spike Firing Approximation (SFA) [11], a framework combining integer training with spike-driven inference, to optimize neuronal firing patterns.

We postulate that information transmission between spiking neurons is governed by spike firing rates. The firing rate of a spiking neuron is operationally defined as:

$$a^l = \frac{1}{T} \sum_{t=1}^{T} \mathbf{S}^l[t] \tag{16}$$

where $a^l$ denotes the spike firing rate, and $\mathbf{S}^l[t]_T$ represents the spike train sequence at layer l over T discrete time steps.

During training, we implement integer activation training by approximating $a^l$ with integer values. This is achieved by replacing the temporal summation $\sum_{t=1}^{T} \mathbf{S}^l[t]$ with single-timestep integer-valued activations:

$$\mathbf{S}_T^l = \lfloor \text{clip}\{\mathbf{x}^l, 0, T\} \rceil \tag{17}$$

where $\mathbf{S}_T^l$ denotes the integer activation within [0,T], with T representing the maximum activation threshold. And clip$\{\mathbf{x}^l, 0, T\}$constrains $\mathbf{x}^l$ to the interval [0,T], followed by rounding to the nearest integer:

In inference, we perform spike activation inference:

$$\mathbf{S}_T^l = \sum_{t=1}^{T} \hat{\mathbf{S}}^l[t] \tag{18}$$

where $\hat{\mathbf{S}}^l[t]$ denotes the spike train sequence in SFA (Spike Firing Approximation) inference. The spatial input to spiking neurons in layer (l+1) is computed as:

$$\mathbf{X}^{l+1} = W^{l+1}a^l = W^{l+1}\frac{1}{T}\mathbf{S}_T^l = \left(\frac{1}{T}\mathbf{W}^{l+1}\right) \sum_{t=1}^{T} \hat{\mathbf{S}}^l[t] \tag{19}$$

The spike sequence $\hat{\mathbf{S}}^l[t]$ consists solely of 0s and 1s, enabling all MAC (Multiply-Accumulate) operations to be converted into sparse AC (Accumulation Operations), thereby ensuring spike-driven computation during inference.

## B.2  Input Representation

In this work, the input consists of two complementary modalities: frame-rate-fixed image frames I and a continuous event stream E(x, y, t, p), where (x, y) denotes pixel coordinates, t represents timestamps, and p indicates polarity. To adapt to deep learning architectures, event streams are typically converted into image-like representations [34, 60, 61]. Compared with complex event representation methods in single-object tracking (such as event voxels [62] and GTP [16]), we employ a simple Time-Integrated Image of Events [34] to encode the event stream within an accumulated temporal window t. This approach maps events into a 3D tensor while preserving temporal information and polarity, ensuring compatibility with deep learning methods and facilitating fusion with existing image frames modalities. Specifically, the conversion process first partitions the event stream along the temporal axis into a set of B bins. The polarity values of events within each bin are accumulated and normalized

to the [0, 255] range using an activation function. The transformation is formulated as follows:

$$t_i^* = \left\lceil B \cdot \frac{t_i - t_1}{t_{N_e} - t_1} \right\rceil \tag{20}$$

$$S(x, y, t) = \sigma\left(\sum_i p_i \, \delta(x - x_i)\delta(y - y_i)\delta(t - t_i^*)\right) \tag{21}$$

$$\sigma(a) = \frac{255}{1 + e^{-\frac{a}{2}}} \tag{22}$$

Where $\sigma(\cdot)$ and $\delta(\cdot)$ represent the activation function and Dirac delta function, respectively. $t_{N_e}$ denotes the total number of events. The generated event time image $S \in \mathbb{R}^{B \times H \times W}$ and the corresponding image frames are processed using standard methods to extract template patches and search patches from both modalities. These patches are then jointly fed into the SpikeFET model.

## C   More Implementation Details

### C.1   Response Loss

We adopt the weighted focal loss [38] for by constraining the similarity. Specifically, we denote the ground truth target center and the corresponding low-resolution equivalent as as $\hat{p}$ and $\bar{p} = [\bar{p}_x, \bar{p}_y]$, respectively. The Gaussian kernel is then applied to generate the groundtruth heatmap: $\mathbf{P}_{xy} = \exp\left(-\frac{(x - \bar{p}_x)^2 + (y - \bar{p}_y)^2}{2\delta_p^2}\right)$, where $\delta_p$ represents the object size-adaptive standard deviation [38]. Thus, the Gaussian-Weighted Focal (GWF) loss function is formulated as:

$$\mathcal{L}_{\text{GWF}} = -\sum_{xy} \begin{cases} (1 - \mathbf{P}_{xy})^\alpha \log(\mathbf{P}_{xy}) & \text{if } \hat{\mathbf{P}}_{xy} = 1 \\ (1 - \hat{\mathbf{P}}_{xy})^\beta (\mathbf{P}_{xy})^\alpha \log(1 - \mathbf{P}_{xy}), & \text{otherwise} \end{cases} \tag{23}$$

where $\alpha$ and $\gamma$ are hyperparameters and are set to 2 and 4, respectively, as suggested in [62]. The response maps of both modalities are normalized by dividing them by a temperature coefficient $\tau$ (empirically set to 2). The final loss function is expressed as: $\mathcal{L}_{\text{Res}} = \mathcal{L}_{\text{GWF}}(\mathbf{R}_F/\tau, \mathbf{R}_E/\tau)$

### C.2   Metrics

When evaluating algorithms in the field of Spiking Neural Networks (SNNs), researchers often adopt theoretical power consumption estimation methods to simplify the analysis of hardware implementation details. Specifically, the energy cost of Artificial Neural Networks (ANNs) is calculated as FLOPs multiplied by $\text{E}_{\text{MAC}}$, while the energy cost of SNNs is determined by FLOPs multiplied by $\text{E}_{\text{AC}}$ and the network spike firing rate. In 45 nm technology, the energy consumption for MAC (Multiply-ACcumulate) and AC (ACcumulate) operations is $\text{E}_{\text{MAC}} = 4.6pJ$ and $\text{E}_{\text{AC}} = 0.9pJ$, respectively [63]. For spiking-based convolutional or multilayer perceptron (MLP) layers, only the additional time step T and the spike firing rate per layer need to be considered. In this paper, we can calculate the peak emissivity of each layer, so the energy consumption of each layer is FLOPs multiplied by EAC multiplied by the peak emissivity of each layer. The subtle difference is that the network structure will affect the number of additions triggered by a single peak. For example, when using different convolution kernel sizes for matrix multiplication, the energy consumption of the same spiky tensor is different. we calculate the theoretical power consumption using the following formula [64]:

$$\text{E}_{\text{ANN}} = \text{E}_{\text{MAC}} \times (\text{FL}_{\text{Conv}} + \text{FL}_{\text{MLP}}) \tag{24}$$

$$\text{E}_{\text{SNN}} = \text{T} \times \text{E}_{\text{AC}} \times \left(\sum_{m=1}^{M} \text{R}_{\text{C}}^{(m)} \times \text{FL}_{\text{Conv}}^{(m)} + \sum_{n=1}^{N} \text{R}_{\text{M}}^{(n)} \times \text{FL}_{\text{MLP}}^{(n)}\right) + \text{T} \times \text{E}_{\text{MAC}} \times \text{FL}_{\text{Conv}} \tag{25}$$

where T denotes the time step length; $\text{R}_{\text{C}}^{(m)}$ and $\text{R}_{\text{M}}^{(n)}$ represent the spike firing rates of the m-th convolutional layer and the n-th fully connected (MLP) layer, respectively, defined as the ratio of non-zero elements in the spike tensor. $\text{FL}_{\text{Conv}}^{m}$ and $\text{FL}_{\text{MLP}}^{(n)}$ correspond to the FLOPs of their respective layers, and $\text{FL}_{\text{Conv}}$ represents the FLOPs of the first and last convolutional layer in the tracking head.

More specifically, the FLOPs of the m-th Conv layer in ANNs are:

$$\text{FL}_{\text{Conv}} = (k_m)^2 \cdot h_m \cdot w_m \cdot c_{m-1} \cdot C_m \tag{26}$$

Where $k_m$ is the kernel size, $(h_m, w_m)$ is the output feature map size, $c_{m-1}$ and $c_m$ are the input and output channel numbers, respectively. The FLOPs of the n-th MLP layer in artificial neural networks is:

$$\text{FL}_{\text{MLP}} = i_n \cdot o_n \tag{27}$$

where $i_n$ and $o_n$ are the input and output dimensions of the MLP layer, respectively. In order to provide readers with a clear understanding of spiking firing rate, we have provided a detailed spiking firing rates of a SpikeFET model in Tab. 11.

## D  Detailed configuration and hyperparameters of SpikeFET models and training

On the Frame-Event tracking benchmark, we used two scales of SpikeFET in Tab. 5 and trained the model in our paper using the hyperparameters in Tab. 6.

Table 5: Configuration of backbone models for different SpikeFET (considering only single branch)

| stage | # Tokens | Layer Specification | | | Tiny | Base |
|---|---|---|---|---|---|---|
| 1 | $\frac{H}{2} \times \frac{W}{2}$ | Downsampling | | Dim | 32 | 64 |
| | | ConvFormer Spike Block | SepSpikeConv | MLP ratio | 2 | |
| | | | Channel Conv | Conv ratio | 4 | |
| | | # Blocks | | | 1 | |
| 2 | $\frac{H}{4} \times \frac{W}{4}$ | Downsampling | | Dim | 64 | 128 |
| | | ConvFormer Spike Block | SepSpikeConv | MLP ratio | 2 | |
| | | | Channel Conv | Conv ratio | 4 | |
| | | # Blocks | | | 1 | |
| 3 | $\frac{H}{8} \times \frac{W}{8}$ | Downsampling | | Dim | 128 | 256 |
| | | ConvFormer Spike Block | SepSpikeConv | MLP ratio | 2 | |
| | | | Channel Conv | Conv ratio | 4 | |
| | | # Blocks | | | 2 | |
| 4 | $\frac{H}{16} \times \frac{W}{16}$ | Downsampling | | Dim | 256 | 512 |
| | | TransFormer Spike Block | SepSpikeConv | MLP ratio | 2 | |
| | | | CSWin-SSA | Gamma ratio | 4 | |
| | | | Channel MLP | MLP ratio | 4 | |
| | | # Blocks | | | 6 | 9 |
| 5 | $\frac{H}{16} \times \frac{W}{16}$ | Downsampling | | Dim | 320 | |
| | | TransFormer Spike Block | SepSpikeConv | MLP ratio | 2 | |
| | | | CSWin-SSA | Gamma ratio | 4 | |
| | | | Channel MLP | MLP ratio | 4 | |
| | | # Blocks | | | 2 | 3 |

Table 6: Hyper-parameters for training on SpikeFET

| Hyper-parameter | Finetune | | | Directly Training |
|---|---|---|---|---|
| | Dual-modal | | Single-modal | Single-modal |
| Traning | Tiny | Base | Tiny | Base |
| Model size | Tiny | Base | Tiny | Base |
| Timestemp | 4 | 8 | 4 | 8 |
| Epochs | 50 | 50 | 50 | 300 |
| Batch size | 32 | 16 | 80 | 40 |
| Optimizer | | | ADAMW | |
| Learning rate | 6e-4 | 7.5e-5 | 6e-4 | 7.5e-4 |
| Learning rate decay | | | Cosine | |
| Warmup eopchs | 5 | 5 | 5 | 20 |
| Weight decay | | | 0.05 | |

# E  CSWin-SSA

CSWin Transformer [36] is an efficient and effective Transformer based general visual task backbone, which we have extended to pulse based Cross-Shaped Windows Spiking Self-Attention (CSWin-SSA). The CSWin-SSA first generates key ($\mathbf{K_S}$), query ($\mathbf{Q_S}$), and value ($\mathbf{V_S}$) vectors by linearly transforming the input $\mathbf{U}$. Then, it performs a $\gamma$-fold channel expansion on $\mathbf{V_S}$ to enhance representation. Next, the CSWinSSA operator applies cross-modal self-attention to establish dynamic associations between template and search region features, enabling adaptive extraction of target features. CSWinSSA is achieved by performing self-attention within the horizontal and vertical parallel stripes that form a cross-shaped window. According to the multihead self-attention mechanism, the input feature $\mathbf{U} \in R^{T \times H \times W \times C}$ will be first linearly projected to $K$ heads, and then each head will perform local self-attention within either the horizontal or vertical stripes. Specifically, the CSWin-SSA module can be formulated as:

$$\mathbf{Q_S} = \mathrm{SN}(\mathrm{Linear}(\mathbf{U})), \ \mathbf{K_S} = \mathrm{SN}(\mathrm{Linear}(\mathbf{U})), \ \mathbf{V_S} = \mathrm{SN}(\mathrm{Linear}_\gamma(\mathbf{U})) \tag{28}$$

$$\mathbf{Q_S} = [\mathbf{Q_S^1}, \mathbf{Q_S^2}, \ldots, \mathbf{Q_S^M}], \mathbf{K_S} = [\mathbf{K_S^1}, \mathbf{K_S^2}, \ldots, \mathbf{K_S^M}], \mathbf{V_S} = [\mathbf{V_S^1}, \mathbf{V_S^2}, \ldots, \mathbf{V_S^M}] \tag{29}$$

$$\mathbf{Y}_k^i = \mathrm{SSA}(\mathbf{Q_S}^{ik}, \mathbf{K_S}^{ik}, \mathbf{V_S}^{ik}) \tag{30}$$

$$\text{H-SSA}_k(X) = [\mathbf{Y}_k^1, \mathbf{Y}_k^2, \ldots, \mathbf{Y}_k^M] \tag{31}$$

$$\mathrm{SSA}(\mathbf{Q_S}, \mathbf{K_S}, \mathbf{V_S}) = \mathrm{SN}(\mathbf{Q_S}\mathbf{K_S}^\top \mathbf{V_S} * \mathrm{scale}) \tag{32}$$

where $\mathbf{Q_S}^i \in R^{(sw \times W) \times C}$ and $M = H/sw$, $i = 1, \ldots, M$. The vertical stripes self-attention can be similarly derived, and its output for $k^{th}$ head is denoted as V-SSA$_k(X)$, K, V are the same. To address the challenge of large values generated by matrix multiplication, a scaling factor (scale) is introduced to regulate the magnitude of the results.

Assuming that the natural image has no directional deviation, we divide the $K$ heads into two parallel groups on average (each group has $K/2$ heads, $K$ is usually an even number). The first group of heads performs horizontal stripe self attention, while the second group of heads performs vertical stripe self attention. Finally, the outputs of these two parallel groups will be reconnected together.

$$\mathbf{U}' = \mathrm{Linear}_{\frac{1}{\gamma}}(\mathrm{Concat}(\text{head}_1, \ldots, \text{head}_K)) \tag{33}$$

$$\text{head}_k = \begin{cases} \text{H-SSA}_k(X), & k = 1, \ldots, K/2 \\ \text{V-SSA}_k(X), & k = K/2 + 1, \ldots, K \end{cases} \tag{34}$$

# F  Experiments of the hyper-parameters

In the proposed SpikeFET framework, the hyper-parameters involved are $\mathcal{L}_{\text{res}}$ weight $\alpha$ and $\mathcal{L}_{\text{sim}}$ weight $\beta$, for which we only present ablation studies around the optimal parameters.

**The ablation study of $\alpha$**  We introduced the ablation study of $\alpha$ in Tab. 7. When $\alpha$ is greater than 1, the fusion response map of the two modalities overly depends on the worst modality, leading to a decrease in metrics, whereas when $\alpha$ is less than 1, the response maps of image frames and event frames cannot be adequately aligned. Therefore, we select $\alpha = 1$.

Table 7: The ablation study of $\alpha$.

|          | $\alpha = 0.5$ | $\alpha = 1$ | $\alpha = 2$ |
|----------|------|------|------|
| AUC(%)   | 56.3 | 56.8 | 55.8 |

**The ablation study of $\beta$**  The ablation study of $\beta$ is shown in Tab. 8. When $\beta = 0.5$, we achieved the best performance. When $\beta$ is too large, excessive emphasis on consistency between two template frames can lead to different features tending towards consistency, resulting in errors. When $\beta$ is too small, it cannot fully make the two template frames consistent in time and space, and the effect is not significant enough.

Table 8: The ablation study of $\beta$.

|  | $\beta = 0.1$ | $\beta = 0.5$ | $\beta = 1$ |
|---|---|---|---|
| AUC(%) | 56.4 | 56.8 | 56.2 |

# G  Runtime and FLOPs

Here we present the specific values of MAC (Multiply-ACcumulate) and AC (ACcumulate), along with a runtime comparison between SpikeFET/SpikeET and other methods on an RTX 4090 GPU for reference. Results for SpikeFET and SpikeET are shown in Tab. 9 and Tab. 10, respectively.

Table 9: Comparison of MAC, AC, and runtime between SpikeFET and other methods.

| Methods | Architecture | Speed (FPS) | MAC/AC | FE108 [8] | | VisEvent [6] | | COESOT [7] | |
|---|---|---|---|---|---|---|---|---|---|
| | | | | AUC(%) | PR(%) | AUC(%) | PR(%) | AUC(%) | PR(%) |
| ANN | DiMP50∗ [45] | - | - | 57.1 | 85.1 | 47.8 | 67.0 | 58.9 | 67.1 |
| | PrDiMP50∗ [46] | - | - | 59.0 | 87.7 | 47.6 | 65.3 | 57.9 | 69.6 |
| | SiamRCNN∗ [47] | - | - | - | - | 49.9 | 65.9 | 60.9 | 71.0 |
| | TrDiMP50∗ [20] | - | - | 60.3 | 91.2 | - | - | 60.1 | 72.2 |
| | TransT50∗ [2] | - | - | 63.9 | 93.0 | 47.4 | 65.0 | 60.5 | 72.4 |
| | ToMP101∗ [48] | - | - | 61.8 | 91.1 | - | - | 59.9 | 67.2 |
| | FENet [8] | 92.45 | 57 | 63.1 | 91.8 | - | - | - | - |
| | OSTrack [3] | 113.29 | 57.02 | - | - | 53.4 | 69.5 | 59.0 | 70.7 |
| | CEUTrack [7] | 116.53 | 57.74 | 55.6 | 84.5 | 53.1 | 69.1 | 62.7 | 76.0 |
| | HRCEUTrack [22] | 123.38 | 52.13 | - | - | - | - | 63.2 | 71.9 |
| | HRMonTrack [22] | - | - | **68.5** | **96.2** | - | - | - | - |
| | ViPT† [25] | **135** | **29.33** | 65.2 | 92.1 | **59.2** | **75.8** | **65.7** | **73.9** |
| | SDSTrack† [19] | 42.03 | 156.33 | **65.8** | 92.6 | **59.7** | **76.7** | 63.7 | 71.7 |
| ANN-SNN | MMHT [15] | 85.48 | 34.98 | 63.0 | 93.6 | 55.1 | 73.3 | **65.8** | 74.0 |
| SNN | **SpikeFET-Tiny** | 106 | 30.34 | **68.5** | **96.5** | 56.8 | 73.5 | 64.0 | **77.9** |
| | **SpikeFET-Base**† | 49.26 | 121.24 | **68.7** | **97.0** | **59.0** | **75.3** | **68.5** | **81.7** |

Table 10: Comparison of MAC, AC, and runtime between SpikeET and other methods.

| Methods | Architecture | Speed (FPS) | MAC/AC | FE108 [8] | | VisEvent [6] | |
|---|---|---|---|---|---|---|---|
| | | | | AUC(%) | PR(%) | AUC(%) | PR(%) |
| ANN | DiMP50 [45] | 71.93 | 55.73 | - | - | 31.5 | 44.2 |
| | PrDiMP50 [46] | 71.82 | 55.73 | - | - | 32.2 | 46.9 |
| | ATOM [49] | 121.11 | 44.321 | - | - | 28.6 | 47.4 |
| | SiamRPN [50] | **518.55** | 6.57 | - | - | 24.7 | 38.4 |
| | STARK [4] | 172.19 | 12.8 | 57.4 | 89.2 | 34.1 | 46.8 |
| | SimTrack [51] | 266.25 | 20.4 | 56.7 | 88.3 | 34.6 | 47.6 |
| | OSTrack [3] | 204.38 | 21.5 | 54.6 | 87.1 | 32.7 | 46.4 |
| | ARTrack [52] | 97.47 | 38 | 56.6 | 88.5 | 33.0 | 43.8 |
| | SeqTrack [53] | 113.09 | 65.8 | 53.5 | 85.5 | 28.6 | 43.3 |
| | HiT [54] | 437 | **4.3** | 55.9 | 88.5 | 34.6 | 47.6 |
| | GRM [55] | 153.18 | 30.9 | 56.8 | 89.3 | 33.4 | 47.7 |
| | HIPTrack [56] | 110.23 | 66.9 | 50.8 | 81.0 | 32.1 | 45.2 |
| ANN-SNN | STNet [13] | 163.40 | - | **58.5** | **89.6** | 35.0 | 50.3 |
| | SNNTrack [14] | - | 9.17 | - | - | **35.4** | **50.4** |
| SNN | SDTrack [16] | - | 30.52 | **66.56** | **91.5** | **37.4** | **51.5** |
| | **SpikeET** | 185.87 | 9.78 | **64.4** | **94.7** | **39.4** | **54.0** |

# H  Impact Statement

This paper proposes a fully spike-based frame-event tracking framework, and demonstrates the effectiveness and necessity of SNNs in target tracking through extensive experimental validation.

This work provides a novel approach and establishes a baseline for achieving low-power, high-performance, and robust visual object tracking, aiming to inspire further research and development in energy-efficient and high-performance tracking systems. At the same time, it provides a new paradigm for developing low-power edge vision computing, demonstrating potential applications in real-time perception fields such as autonomous driving, and promoting the engineering process of neural morphological computing technology. However, the misuse of Object tracking technology can have a negative impact on personal privacy.

# I  Limitations

Although SpikeFET has achieved commendable accuracy and robustness in frame-event tracking through SNNs, its sparse characteristic results in insufficient pre-training on tracking datasets, leading to suboptimal performance on VisEvent [6] compared to state-of-the-art trackers. Meanwhile, deploying the tracker on edge computing platforms (e.g., brain-inspired chips or neuromorphic chips) could further exploit SNNs' low-power characteristics. These challenges will constitute critical research directions for future exploration.

Table 11: Layer spiking firing rates of model SpikeFET-Tiny on VisEvent.

| | | | | Image | Event |
|---|---|---|---|---|---|
| | Downsampling | | Conv | 1 | 1 |
| Stage 1 | ConvFormer Spike Block | SepSpikeConv | PWConv1 | 0.3645 | 0.2723 |
| | | | DWConv | 0.3314 | 0.3330 |
| | | | PWConv2 | 0.3526 | 0.3703 |
| | | Channel Conv | Conv1 | 0.4293 | 0.3656 |
| | | | Conv2 | 0.0740 | 0.0771 |
| Stage 2 | Downsampling | | Conv | 0.3203 | 0.2694 |
| | ConvFormer Spike Block | SepSpikeConv | PWConv1 | 0.1791 | 0.1616 |
| | | | DWConv | 0.2381 | 0.2243 |
| | | | PWConv2 | 0.1712 | 0.1702 |
| | | Channel Conv | Conv1 | 0.2680 | 0.2442 |
| | | | Conv2 | 0.0371 | 0.0302 |
| Stage 3 | Downsampling | | Conv | 0.2245 | 0.2031 |
| | ConvFormer Spike Block | SepSpikeConv | PWConv1 | 0.1876 | 0.1719 |
| | | | DWConv | 0.2484 | 0.2379 |
| | | | PWConv2 | 0.1300 | 0.1248 |
| | | Channel Conv | Conv1 | 0.1818 | 0.1721 |
| | | | Conv2 | 0.0197 | 0.0175 |
| | ConvFormer Spike Block | SepSpikeConv | PWConv1 | 0.2642 | 0.2465 |
| | | | DWConv | 0.1486 | 0.1412 |
| | | | PWConv2 | 0.1160 | 0.1102 |
| | | Channel Conv | Conv1 | 0.1960 | 0.1874 |
| | | | Conv2 | 0.0133 | 0.0116 |
| stage4 | Downsampling | | Conv | 0.2335 | 0.2050 |
| | TransFormer Spike Block1 | SepSpikeConv Conv-1/2/3 | | 0.1408 | |
| | | CSWin-SSA | $Q_S$ | 0.2372 | |
| | | | $K_S$ | 0.1077 | |
| | | | $V_S$ | 0.1851 | |
| | | | $Q_S(K_S^T V_S)$ | 0.5644 | |
| | | | Linear | 0.7385 | |
| | | Channel MLP | Linear 1 | 0.2194 | |
| | | | Linear 2 | 0.0156 | |
| | TransFormer Spike Block1 | SepSpikeConv Conv-1/2/3 | | 0.1811 | |
| | | CSWin-SSA | $Q_S$ | 0.1901 | |
| | | | $K_S$ | 0.0584 | |
| | | | $V_S$ | 0.0939 | |
| | | | $Q_S(K_S^T V_S)$ | 0.1180 | |
| | | | Linear | 0.5973 | |
| | | Channel MLP | Linear 1 | 0.2867 | |
| | | | Linear 2 | 0.0150 | |
| | TransFormer Spike Block2 | SepSpikeConv Conv-1/2/3 | | 0.1782 | |
| | | CSWin-SSA | $Q_S$ | 0.1834 | |
| | | | $K_S$ | 0.0460 | |
| | | | $V_S$ | 0.1147 | |
| | | | $Q_S(K_S^T V_S)$ | 0.1160 | |
| | | | Linear | 0.5349 | |
| | | Channel MLP | Linear 1 | 0.2977 | |
| | | | Linear 2 | 0.0140 | |

| | | | | Image | Event |
|---|---|---|---|---|---|
| | | SepSpikeConv Conv-1/2/3 | | 0.1816 | |
| | TransFormer Spike Block3 | CSWin-SSA | $Q_S$ | 0.1633 | |
| | | | $K_S$ | 0.0532 | |
| | | | $V_S$ | 0.0909 | |
| | | | $Q_S(K_S^T V_S)$ | 0.0962 | |
| | | | Linear | 0.4846 | |
| | | Channel MLP | Linear 1 | 0.2922 | |
| | | | Linear 2 | 0.0103 | |
| | | SepSpikeConv Conv-1/2/3 | | 0.1733 | |
| | TransFormer Spike Block4 | CSWin-SSA | $Q_S$ | 0.1925 | |
| | | | $K_S$ | 0.0422 | |
| | | | $V_S$ | 0.0808 | |
| | | | $Q_S(K_S^T V_S)$ | 0.0676 | |
| | | | Linear | 0.3879 | |
| | | Channel MLP | Linear 1 | 0.2574 | |
| | | | Linear 2 | 0.0116 | |
| | | SepSpikeConv Conv-1/2/3 | | 0.1558 | |
| | TransFormer Spike Block5 | CSWin-SSA | $Q_S$ | 0.1886 | |
| | | | $K_S$ | 0.0532 | |
| | | | $V_S$ | 0.1221 | |
| | | | $Q_S(K_S^T V_S)$ | 0.1294 | |
| | | | Linear | 0.5527 | |
| | | Channel MLP | Linear 1 | 0.1854 | |
| | | | Linear 2 | 0.0146 | |
| | Downsampling | Conv | | 0.1564 | |
| | | SepSpikeConv Conv-1/2/3 | | 0.1000 | |
| | TransFormer Spike Block1 | CSWin-SSA | $Q_S$ | 0.1951 | |
| | | | $K_S$ | 0.0369 | |
| | | | $V_S$ | 0.0487 | |
| | | | $Q_S(K_S^T V_S)$ | 0.0449 | |
| | | | Linear | 0.3142 | |
| | | Channel MLP | Linear 1 | 0.2651 | |
| | | | Linear 2 | 0.0032 | |
| stage5 | | SepSpikeConv Conv-1/2/3 | | 0.0900 | |
| | TransFormer Spike Block2 | CSWin-SSA | $Q_S$ | 0.1978 | |
| | | | $K_S$ | 0.0165 | |
| | | | $V_S$ | 0.0073 | |
| | | | $Q_S(K_S^T V_S)$ | 0.0148 | |
| | | | Linear | 0.1635 | |
| | | Channel MLP | Linear 1 | 0.1314 | |
| | | | Linear 2 | 0.0054 | |
| | | Conv1 | | 0.1519 | 0.1511 |
| | | Conv2 | | 0.0648 | 0.0665 |
| | Ctr | Conv3 | | 0.0800 | 0.0752 |
| | | Conv4 | | 0.1429 | 0.1439 |
| | | Conv5 | | 1 | 1 |
| | | Conv1 | | 0.1519 | 0.1511 |
| | | Conv2 | | 0.0818 | 0.0865 |
| Tracking | Offset | Conv3 | | 0.0912 | 0.0844 |
| | | Conv4 | | 0.1273 | 0.1213 |

|      |       | Image  | Event  |
| ---- | ----- | ------ | ------ |
|      | Conv5 | 1      | 1      |
|      | Conv1 | 0.1511 | 0.1511 |
|      | Conv2 | 0.0818 | 0.0865 |
| Size | Conv3 | 0.1126 | 0.1112 |
|      | Conv4 | 0.1526 | 0.1531 |
|      | Conv5 | 1      | 1      |

