# OpenReview forum: "Fully Spiking Neural Networks for Unified Frame-Event Object Tracking"
_NeurIPS.cc/2025/Conference — NeurIPS 2025 poster_

### Official Review · Reviewer_VsCN · 2025-06-30

**Clarity:** 3
**Significance:** 3
**Originality:** 3
**Rating:** 4
**Confidence:** 4

**Summary:**

This paper proposes a fully SNN-based framework for multimodal tracking with frame and event data. The architecture comprises a ConvFormer Spike Block for modality-specific feature extraction and a TransFormer Spike Block for feature fusion. The core contributions are: (1) the Randomized Patchwork Module (RPM), which mitigates convolution-induced positional bias via stochastic spatial reordering; and (2) a Spatial-Temporal Regularization (STR) strategy that enhances the consistency of dual-template features in latent space. Experiments on multiple datasets demonstrate the effectiveness of the proposed method.

**Questions:**

See above Weaknesses

**Ethical Concerns:**

["NO or VERY MINOR ethics concerns only"]

**Final Justification:**

The authors’ responses have addressed most of my concerns, and I have accordingly increased my score. However, the reported runtime performance still requires more detailed information, as there appear to be discrepancies between the speeds presented in this work and those reported in the original papers.

**Limitations:**

yes

**Quality:**

3

**Strengths And Weaknesses:**

**Strengths:**

•  The exploration of a fully SNN-based framework for event-driven tracking is commendable, as most existing methods rely on hybrid ANN-SNN designs.

•  The proposed method achieves performance on par with ANN-based models, highlighting the effectiveness of the fully SNN architecture.

**Weaknesses:**

•  The model is pretrained on large-scale RGB datasets such as COCO [40], LaSOT [41], TrackingNet [42], and GOT-10K. However, the fairness of comparison with other methods is unclear.

•  The proposed RPM and SDTrack modules treat the search and template frames similarly. Moreover, RPM is designed to mitigate the degradation of translation invariance caused by padding, yet it is only applied in one direction. It remains unclear whether padding artifacts persist in the other directions.

•  Besides parameter count, inference speed is also a critical metric for tracking tasks. Please provide a runtime comparison with other methods.

•  The paper does not specify the timestamp configuration used in the experiments.

•  As shown in Table 3, the baseline performs poorly while the addition of RPM leads to a large performance gain. Such a dramatic improvement warrants further explanation and justification.

---

> ### Author Rebuttal · Authors · 2025-07-31
>
> We thank reviewer VsCN for the valuable and constructive comments. We address the concerns as follows.
> ## W1: The model is pretrained on large-scale RGB datasets such as COCO [40], LaSOT [41], TrackingNet [42], and GOT-10K. However, the fairness of comparison with other methods is unclear.
> Thanks for pointing out the misunderstanding regarding Table 1. We clarify that to ensure a fair comparison with pre-training methods such as ViPT [25] and SDSTrack [19], we followed their approach by pre-training SpikeFET-Base on COCO [40], LaSOT [41], TrackingNet [42], and GOT-10K [43]. The results demonstrate that SpikeFET-Base outperforms both methods on FE108 [8] and COESOT [7]. In contrast, to compare with most non-pre-trained baselines, SpikeFET-Tiny and SpikeET were not pre-trained on these datasets. Their results also show superior performance over the majority of non-pretrained methods, except for MMHT's [15] AUC performance on COESOT [7]. We will explicitly highlight this distinction in the revised table and corresponding textual description.
> ## W2: The proposed RPM and SDTrack modules treat the search and template frames similarly. Moreover, RPM is designed to mitigate the degradation of translation invariance caused by padding, yet it is only applied in one direction. It remains unclear whether padding artifacts persist in the other directions.
> We thank the reviewer for the comment. We clarify that our proposed RPM is fundamentally different from the Intrinsic Position Learning (IPL) introduced in SDTrack [19]. First, IPL is motivated by the idea that a joint positional embedding can enhance the cross-correlation performance; its goal is to effectively learn positional information. In contrast, RPM stems from the observation that the padding in convolutional layers introduces a systematic bias during tracking and degrades translation invariance; its purpose is to mitigate this degradation. Second, IPL adds redundant padding information to the input image, which hurts SDTrack's [19] performance, whereas RPM actively reduces such redundancy and further boosts accuracy. Finally, although at inference RPM randomly selects only one direction, during training it considers all four possible placements—top, bottom, left, and right—of the search region within the padded image, thereby eliminating the biasing effect of padding in the unused directions.
> ## W3: Besides parameter count, inference speed is also a critical metric for tracking tasks. Please provide a runtime comparison with other methods.
> We clarify that runtime is not a primary metric considered in this work, as **it can be reduced through hardware acceleration.** However, **based on suggestions, we have added a comparison of the runtime between SpikeFET/SpikeET and other methods on an RTX 4090 GPU.** For certain baselines of SpikeFET, such as DiMP50 [44], PrDiMP50 [45], SiamRCNN [46], TrDiMP50 [20], TransT50 [2], and ToMP101 [47], we have referenced the AUC metrics from other papers. This is because these methods, originally frame-based trackers, were extended to frame-event fusion trackers using early fusion approaches. Due to our lack of detailed understanding of their specific implementations, we were unable to replicate their algorithms and thus could not accurately calculate their inference times. The overall results are presented in Rebuttal Table A. Specifically: our SpikeFET-Base inference time outperforms those of SDSTrack [19] and HRMonTrack [22]. Beyond this, the inference time of SpikeFET-Tiny is comparable to other methods and falls within an acceptable range. Additionally, our SNN algorithm can be deployed on neuromorphic chips in the future to further reduce inference time.
> **Rebuttal Table A.** Comparison of inference time between SpikeFET and other baselines, with inference time tested on RTX 4090 using the same testing methodology as OSTrack [3].
> |Method|FENet [8]|OSTrack [3]|CEUTrack [7]|HRCEUTrack [22]|HRMonTrack [22]|ViPT [25]|SDSTrack [19]|MMHT [15]|SpikeFET-Tiny|SpikeFET-Base|
> |-|-|-|-|-|-|-|-|-|-|-|
> |**Inference Speed (FPS)**|92.45|127.8|116.53|123.38|39.7|135.43|42.03|85.48|106.23|49.26|
>
> For SpikeET, we tested all methods except SNNTrack [14] and SDTrack [16] due to their non-public implementations. The overall results are presented in Rebuttal Table B. Specifically: our SpikeET inference time outperforms most networks and remains comparable to other methods within an acceptable range. Additionally, our SNN algorithm can be deployed on neuromorphic chips in the future to further reduce inference time. We appreciate your emphasis on these critical points. All testable methods will be included in the revised version for inference speed evaluation.
> **Rebuttal Table B.** Comparison of inference time between SpikeET and other baselines, with inference time tested on RTX 4090 using the same testing methodology as OSTrack [3].
> |Method|DiMP50 [44]|PrDiMP50 [45]|ATOM  [48]|SiamRPN [49]|STARK [4]|SimTrack [50]|OSTrack [3]|
> |-|-|-|-|-|-|-|-|
> |**Inference Speed (FPS)**|71.93|72.82|121.11|518.55|172.19|266.25|204.38|
> |**Method**|**ARTrack [51]**|**SeqTrack [52]**|**HiT [53]**|**GRM [54]**|**HIPTrack[55]**|**STNet [13]**|**SpikeET**|
> |**Inference Speed (FPS)**|97.4|113.09|437|153.18|110.23|163.40|185.87|
> ## W4: The paper does not specify the timestamp configuration used in the experiments.
> As detailed in Appendix A.2, we convert the event stream into an event frame by uniformly partitioning a temporal slice of the stream into B = 3 segments and mapping the events in each segment into one of the three RGB channels. On COESOT [7] we employ a 40 ms slice for each segment, while on FE108 [8] and VisEvent [6] we instead collect the events that fall between the timestamps of two consecutive image frames, split that interval into three equal sub-intervals, and map them to the RGB channels; the resulting event frames are used for training and inference.
> ## W5: As shown in Table 3, the baseline performs poorly while the addition of RPM leads to a large performance gain. Such a dramatic improvement warrants further explanation and justification.
> We thank the reviewer for this insightful observation; it is precisely the motivation behind our RPM module. We attribute the observed phenomenon to two key factors.
>
> First, padding severely undermines translation invariance and thereby degrades performance, a problem that has already been noted in works such as SiamRPN++ [17] and SiamDW [18], whose accuracies increased markedly once padding effects were mitigated. With sparse event data—where informative content is already limited—the detrimental impact of padding becomes even more pronounced. Consequently, our RPM module yields larger accuracy gains.
>
> Second, FE108 [8] outperforms VisEvent [6] by a wider margin because FE108 [8] contains not only simple scenes (stationary camera, moving object in a fixed environment) but also challenging conditions such as high-exposure and low-light sequences. Our SNN-based SpikeFET excels at leveraging the sparse event information in these challenging settings, leading to larger improvements on FE108 [8] than on VisEvent [6].

---

> > ### Comment · Reviewer_VsCN · 2025-08-09
> >
> > I appreciate the authors’ responses, which have addressed most of my concerns.

---

> > > ### Author Response · Authors · 2025-08-09
> > >
> > > Thank you for your feedback. We are pleased to hear that our responses have addressed most of your concerns, and we sincerely appreciate your recognition of our work.

---

### Official Review · Reviewer_dnLE · 2025-07-02

**Clarity:** 2
**Significance:** 2
**Originality:** 3
**Rating:** 4
**Confidence:** 5

**Summary:**

The authors present SpikeFET, the first full SSN based object tracker using both events and frame data. The network adds a Random Patchwork Module and Spatial-Temporal Regularization to improve spatial-temporal invariance and consistency. Features are extracting independently for frames and events of image triplets. The features are then fused with a Spiking based transformer. By feeding two samples of the template into the tracking network, end performance can dramatically improve. SpikeFET is extensively tested across 2 benchmarks and compared with a diverse set of baselines. SpikeFET can improve tracking performance while minimizing power use of the system.

**Questions:**

1.	What is the expected processing rate of SpikeFET? How big are the event bins? How much delay is in the SNN between input to output spike?
2.	What is the difference between table 1 and 2? The caption suggests table 2 is on event only input, but at least [44] and [45] are frame-based methods. The authors might consider merging the tables and adding a column indicating the method inputs to be clearer.
3.	Based on the ablation, RPM is the primary component. How much of the improvement is due to simply providing the network with another sample of the template (i.e. more data), than a fundamental insight into model architecture/training processes?
4.	Can the authors comment on parameter scaling SpikeFET? On FE108, it seems 3x the parameters had very little effect, while on COESOT it had a larger effect.
5.	Since frame-based cameras require more energy than event cameras, how does using a fused approach compare to an event only system in terms of overall energy used.

If the authors can clarify these points, I will raise my score.

**Ethical Concerns:**

["NO or VERY MINOR ethics concerns only"]

**Final Justification:**

The authors resolved most of my concerns. Although the paper is not state of the art in all aspects, given its the first fully SNN method for this task, its an interesting direction for future research. Physical testing on neuromorphic chips would have made this a more solid accept.

**Limitations:**

yes

**Paper Formatting Concerns:**

Figure 1 should make it clear the "power" is a theoretical value and not a real world measurement of the systems.

**Quality:**

3

**Strengths And Weaknesses:**

Strengths

-	SpikeFET is a novel method for object tracking with fused event + frame data.
-	SpikeFET has state of the art performance. The authors compare against a variety of other methods including frame only, event only, and frame+event methods by reporting results on 3 different datasets.
-	The method diagrams aide understanding SpikeFET’s flow and how RPM and STR operate.
-	The ablation study effectively compares the major steps in SpikeFET (RPM, Et, STR), and the choices in fusion method.

Weakness

-	The paper only visualizes 2 samples of tracking results, making it difficult to evaluate the method’s success.
-	Inference mode is lacking detail. For instance, how are the 2 template images chosen?
-	Evaluation is theoretical/in simulation only and power usage is missing for some baseline methods.
-	Many important details are relegated to the appendix instead of the main paper (i.e. network input and datasets).
-	Lots of baseline methods are shown, but a description of method differences between SpikeFET and them is lacking.
-	The primary novelty that contributes to improvement over baselines seems to be in passing more data into the network, which is not surprising or significant.

---

> ### Author Rebuttal · Authors · 2025-07-31
>
> We sincerely thank Reviewer dnLE for the constructive and valuable comments. The concerns are addressed as follows.
> ## W1: The paper only visualizes 2 samples of tracking results, making it difficult to evaluate the method’s success.
> Thank you for the reviewers' comments. We have conducted extensive visualizations, and SpikeFET consistently demonstrates strong performance. However, due to space limitations, we only included two figures in the current manuscript. We will incorporate more visualization results into the revised version.
>
> ## W2: Inference mode is lacking detail. For instance, how are the 2 template images chosen?
> We appreciate the reviewers for identifying this limitation, which was an oversight in our work. In our inference process, we did not employ a dynamic template update strategy. Instead, we consistently replicated the first frame to use two template frames (both derived from the first frame) and one search frame for inference. We will provide this clarification after lines 213–214 in the revised manuscript.
> ## W3: Evaluation is theoretical/in simulation only and power usage is missing for some baseline methods.
> We sincerely appreciate the reviewer for highlighting this critical limitation. Currently, most mainstream studies primarily compare theoretical power consumption, lacking practical power measurements. This gap exists because a model's actual power consumption is hardware-dependent. **To ensure fair comparisons across methods, we adopted a widely-used hardware-agnostic approximation method [62, 63] for estimating power consumption in both ANNs and SNNs**. We are now conducting tests on relevant chips and will deploy our SNNs algorithm onto neuromorphic chips for practical power evaluations in future work.
>
> We acknowledge that for certain baselines of SpikeFET—such as DiMP50 [44], PrDiMP50 [45], SiamRCNN [46], TrDiMP50 [20], TransT50 [2], ToMP101 [47], and HRMontrack [22]—we referenced metrics reported in existing literature. This approach was necessary because these methods originally started as frame-based trackers and were extended to frame-event fusion trackers using early fusion approaches. Due to our limited access to implementation specifics, we were unable to reproduce their algorithms or precisely calculate theoretical power consumption. Consequently, we derived an approximated computational result based on our reproduction efforts, which will be supplemented in the revised manuscript. For FENet [8], we obtained a theoretical power consumption of 262.2 mJ, which also exceeds that of SpikeFET-Base.
>
> For certain baselines of SpikeET, such as DiMP50 [44], PrDiMP50 [45], SiamRCNN [46], and ATOM [48], their theoretical power consumption values are detailed in the following table:
>
> |Method|DiMP50 [44]|PrDiMP50 [45]|SiamRCNN [46]|ATOM [48]|
> |-|-|-|-|-|
> |**Power (mJ)**|256.37|258.37|30.199|203.88|
>
> SpikeET **maintains the lowest power consumption**, and we will include this comparative analysis in the revised manuscript.
> ## W4: Many important details are relegated to the appendix instead of the main paper (i.e. network input and datasets).
> We apologize for any confusion caused by placing important details in the appendix due to space constraints; we will reformat and incorporate these details in the revised edition.
> ## W5: Lots of baseline methods are shown, but a description of method differences between SpikeFET and them is lacking.
> We thank the reviewer for this valuable comment. To the best of our knowledge, the proposed SpikeFET is the first fully-spiking frame–event fusion tracker, whereas all baselines—e.g., DiMP50 [44], OSTrack [3], CEUTrack [7], ViPT [25], and SDSTrack [19]—are built upon artificial neural networks (ANNs). MMHT [15], on the other hand, adopts an ANN-SNN hybrid architecture and is therefore not a fully-spiking tracker either. Furthermore, SpikeFET borrows the overall pipeline from CEUTrack but replaces the patch embedding with several convolutional layers, employs two spiking tracking heads, and converts every component into SNNs form. ViPT [25] is likewise built upon OSTrack [3] and treats the event modality as an auxiliary cue, whereas SDSTrack [19] extends OSTrack [3] with a self-distillation scheme. Earlier methods such as DiMP50 [44] and SiamRCNN [46] adopt Siamese architectures. We will add this clarification in the revised manuscript.
> ## W6 and Q3: The primary novelty that contributes to improvement over baselines seems to be in passing more data into the network, which is not surprising or significant.
> We clarify that the performance improvement is not achieved by adding more data. In addition, SNNs currently underperform ANNs in complex computer vision tasks such as detection and tracking. For the first time, we propose a fully-spiking fusion tracking network that achieves performance on par with ANNs — and even surpasses ANNs on some datasets — while consuming significantly less power than ANNs. The analysis is as follows:
>
> In our ablation study (Table 3 #1), we used **six input images** (line 279). This configuration demonstrates that when using only **two template frames and one search frame** per modality without the RPM and adopting conventional input tokens, the model performance significantly deteriorates. Particularly on the FE108 dataset, this setup causes ~50% performance degradation compared to configuration #2.
>
> Additionally, we conducted experiments without RPM using one template frame and one search frame per modality, resulting in four total inputs (two modalities × two inputs each). The results are presented in the table below:
> |Configuration|4-input|6-input|6-input with RPM|
> |-|-|-|-|
> |**AUC (%)**|30.8|32.1|66.8|
> |**PR (%)**|43.9|45.0|94.9|
>
> We observe that without RPM, the six-input configuration yields only marginally higher performance than the four-input setup on FE108 [8]. This conclusively demonstrates that the primary innovation driving baseline improvement is RPM's effective mitigation of padding-induced degradation in translation invariance. In contrast, utilizing additional data provides minimal performance gains relative to RPM's contribution. We sincerely appreciate your emphasis on these important related works. We will incorporate the ablation results for the four-input configuration into the revised manuscript.
> ## Q1: What is the expected processing rate of SpikeFET? How big are the event bins? How much delay is in the SNN between input to output spike?
> (1) We stack the event streams into event frames for training and inference based on the characteristics of each dataset. For COESOT [7], we use a 40ms time window; for FE108 [8] and VisEvent [6], we use the time interval between adjacent image timestamps. Thus, SpikeFET processes event stream data from each 40ms window or the image sampling interval, i.e., the processing rate is 25Hz or matches the image frame rate. In theory, a 10ms window or faster could be used, but to ensure a fair comparison with existing methods, we chose 40ms.
> (3) The end-to-end latency measurements for SpikeFET-Tiny, SpikeFET-Base, and SpikeET on an RTX 4090 GPU are 106 Hz, 49.26 Hz, and 185.87 Hz respectively.
> ## Q2: What is the difference between table 1 and 2? The caption suggests table 2 is on event only input, but at least [44] and [45] are frame-based methods. The authors might consider merging the tables and adding a column indicating the method inputs to be clearer.
> We thank the reviewer for pointing out the error in our paper. We incorrectly listed identical results for DiMP [44] on FE108 [8] in two different tables; this was an oversight on our part. Table 1 reports methods that are trained and evaluated on frame-event fused data, whereas Table 2 reports methods trained and evaluated solely on event data. In Table 2, [44] and [45] denote frame-based models that were trained and tested on event data. After the event-to-frame conversion described in Appendix A.2, the event stream is transformed into event frames, enabling frame-based models to process event data. We appreciate the suggestion and will merge the two tables in the revised version, adding a column to indicate the input modality.
> ## Q4: Can the authors comment on parameter scaling SpikeFET? On FE108, it seems 3x the parameters had very little effect, while on COESOT it had a larger effect.
> As discussed in lines 243–244 of the original manuscript, we believe that the FE108 [8] dataset is both too small and too simple, causing our SpikeFET-Base model to tend toward overfitting. By contrast, COESOT [7] is larger and more challenging—for example, the ratio of training sequences is 76 : 827 in their favor—so COESOT [7] exerted a greater influence. The near-identical performance of Tiny and Base on FE108 [8] is likewise observed in SDTrack [16], which improves AUC by only 0.9 % and PR by merely 0.2 % relative to other datasets, a marginal gain.
> ## Q5: Since frame-based cameras require more energy than event cameras, how does using a fused approach compare to an event only system in terms of overall energy used.
> We clarify that because event images are sparse while frame images are dense, the spike firing rate of event images is lower than that of frame images. This means that frame-based cameras require more energy. Furthermore, fusion methods require two separate branches to process both modalities, resulting in higher power consumption than processing either single modality alone. Therefore, the power consumption of a fusion network is ultimately higher than that of an event-only network in theory, but still lower compared to that of ANNs.
> ## Q6: Figure 1 should make it clear the "power" is a theoretical value and not a real world measurement of the systems.
> Thank you for pointing out this issue. We will provide an accurate definition of this term and make the necessary revisions in the revised manuscript.

---

> > ### Comment · Reviewer_dnLE · 2025-08-02
> >
> > I appreciate the authors' response to my concerns. Does RPM with 4-inputs (one template + one search per modality) yield a similar improvement as the 6-input case? That would help confirm it is indeed an intrinsic property of RPM causing the improvement.

---

> > > ### Author Response · Authors · 2025-08-03
> > >
> > > We sincerely appreciate the reviewer’s feedback. To confirm that the intrinsic properties of RPM are indeed responsible for the observed improvements, we have conducted the following experiments.
> > >
> > > In our experiment, to meet the input requirements of RPM, we replaced the second template frame in each modality with padding, so that only 4 out of the 6 input images contain information. The experimental results are shown in the table below. RPM with 4 inputs does achieve improvements similar to the 6-input setting; however, since the 6-input setting contains more data, the results with 4 inputs are slightly lower than those with 6 inputs. This confirms that the intrinsic property of RPM is indeed the reason for the improvements. In addition, the 6-input setting allows for the further incorporation of STR, as shown in Table 3 of ablation experiments in the main text, thereby achieving better tracking performance.
> > >
> > > |Configuration|4-input|6-input|4-input with RPM|6-input with RPM|
> > > |-|-|-|-|-|
> > > |**AUC (%)**|30.8|32.1|66.1|66.8|
> > > |**PR (%)**|43.9|45.0|93.9|94.9|
> > >
> > > We will include the above experiment in the ablation studies of the revised version, and we believe that these additional experiments, analyses, and explanations can significantly enhance the quality of our submission.

---

> > > > ### Comment · Reviewer_dnLE · 2025-08-03
> > > >
> > > > I appreciate this additional experiment. It does look like with only 4-inputs, the method may be slightly less competitive with the baseline methods, but the authors do acknowledge limitations with SNNs vs ANNs for performance.
> > > >
> > > > I still have some concerns around the power claims and whether they are sufficiently validated. For instance, ViPT seems to have comparable power usage and performance discounting the "significant" power reduction claims.
> > > >
> > > > Also, although the authors attempt to ignore hardware dependence, it appears the $E_{MAC}$ and $E_{AC}$ multipliers are for $45$nm technology (from a decade ago). Perhaps reporting raw FLOP counts would better achieve the stated goal of being hardware-agnostic. As the authors note, various forms of hardware acceleration could impact the real advantage of SNN over ANN which is unexplored. I understand most SNN papers are limited in this regard, so this is not a unique weakness. I may increase my score if the authors can comment on ViPT, and after reviewing the discussions with other reviewers.

---

> ### Author Response · Authors · 2025-08-03
>
> We appreciate the reviewer’s quick feedback. Below, we address the further concerns.
> ### Q1: Only 4-inputs, the method may be slightly less competitive with the baseline methods.
> As previously mentioned, although the performance of the 4-input model is slightly lower than that of the 6-input model, the 4-input model can still achieve state-of-the-art results on COESOT (with the 6-inputs and STR outperforming the second-best method by 2.7%), and it also surpasses most trackers on FE108. Furthermore, the advantages of the 6-input model can be fully leveraged by STR, leading to even better tracking performance.
> ### Q2: ViPT seems to have comparable power usage and performance discounting the "significant" power reduction claims.
> Since ViPT employs a simple early fusion method by directly adding the two modalities, compared to the configuration without addition, its power consumption is halved, and it also consumes less power than other ANN-based methods. Conversely, SpikeFET, constrained by its architecture, cannot use a similar method to ViPT and instead uses two separate branches, which increases power consumption. This is reflected in the ablation study in the main text (Table 4 #5). Furthermore, experiments show that SpikeFET achieves performance comparable to ViPT only on the VisEvent dataset, but achieves significant improvements of 3.5% and 2.8% on the FE108 and COESOT datasets, respectively. This is because SpikeFET is particularly well-suited for handling datasets with complex scenarios (COESOT) and data containing challenging scenes (FE108). In these scenarios, the inherent advantages of event cameras become evident, enabling the SNN-based SpikeFET to demonstrate superior performance.
>
> Additionally, our proposed SpikeFET is the first fully spiking frame-event fusion framework. It elevates the performance of SNN-based tracking to a level comparable with ANN-based tracking while achieving an optimal balance between power consumption and performance—this represents the significance and motivation of our work. However, since SNN-based tracking still lags behind ANN-based tracking to some extent, we position SpikeFET as the first baseline in this field. We firmly believe SpikeFET possesses immense potential, and with further improvements, it can fully surpass ANNs in both performance and power efficiency. Meanwhile, SpikeET also demonstrates outstanding performance in the pure event domain, surpassing all existing event-based trackers, thereby proving the excellence of our entire SNNs framework.
> ### Q3: Perhaps reporting raw FLOP counts would better achieve the stated goal of being hardware-agnostic.
> The reason for not using FLOPs is that the power consumption of addition operations (AC) and multiplication-accumulation operations (MAC) is significantly different. The advantage of SNNs lies in converting dense floating-point multiplications into sparse addition operations through 0/1 spikes. To quantify this advantage, the theoretical power consumption estimation is commonly used in the academic community [11,16,63]. In the future, the actual power consumption will be even lower when deployed on neuromorphic hardware.
>
> Thank you very much for your patience and multiple communications. Your involvement has significantly improved the quality of our work. We believe that these additional analyses and explanations will greatly enhance the quality of our submission.

---

> > ### Comment · Reviewer_dnLE · 2025-08-03
> >
> > I appreciate the back and forth with the authors. I will be following potential discussions with the other reviewers before changing my score.
> >
> > On ViPT, I am not sure the "significant improvement" claims are well validated given there are no error bars, confidence intervals, or statistical tests and the improvement is relatively small. But I think its okay to not overwhelmingly beat the other methods since this is the first full SNN method.
> >
> > On power consumption, it is a shame the community continues using outdated and imperfect analysis. Directly reporting MAC vs AC flops would better achieve the goal of being hardware agnostic. However, there are other non-hardware specific details ignored such as potential quantization and cache friendliness which can also impact power consumption. Given SNNs are sparse and have irregular memory access, its possible the compute cost may be dominated by cache misses ([16] places cache access cost at a least 20x more than an add), but of course with newer neuromorphic chips these characteristics may change. This more rigorous analysis would likely deserve its own paper, but factors into the strength of claims able to be made in this work.

---

> > > ### Author Response · Authors · 2025-08-04
> > >
> > > We appreciate the reviewer's response, but please allow us an opportunity to discuss your feedback further.
> > >
> > > First of all, I would like to thank the reviewer for mentioning and agreeing with the statement, "I think it's okay not to overwhelmingly beat the other methods since this is the first full SNN method." We acknowledge that our use of the term "significant" was misplaced, as it does not appear to be adequately demonstrated on ViPT. We will update our wording accordingly. However, at the very least, SpikeFET has indeed achieved the best balance between power consumption and performance.
> > >
> > > Secondly, we acknowledge the reviewer's points regarding the impact of potential quantization and cache friendliness on power consumption. We recognize that our current analysis only considers the computational costs of matrix multiplication and addition, which is incomplete. We will include a discussion of this aspect in the "limitations" section in the revised version, but this indeed remains a common challenge across the entire field.
> > >
> > > Beyond this, simply comparing MAC and AC floating-point operations is fundamentally unfair to SNNs. The reduction in energy consumption of SNNs stems precisely from the decreased AC computation load due to sparse spike activity, and the actual energy consumption of MAC versus AC operations also differs. Simplistically equating MAC and AC operations for comparison deals a significant blow to the entire SNN field. Consequently, in recent years, many studies [1,2,3,4,5] have used theoretical energy consumption as an approximate metric to quantify this advantage, including the "Theoretical energy consumption evaluation" section on page 15 of the speck chip [4] published in *Nature Communications* last year, which also mentions theoretical power consumption. Although theoretical energy consumption is an imperfect metric, it indeed serves as a convenient and simple evaluation indicator, making SpikeFET a valuable baseline to drive progress across the field. Furthermore, with the rapid development of neuromorphic hardware—such as the gesture recognition-capable speck chip, Tianjic chip, and others—we will also conduct on-chip verification comparing actual power consumption with theoretical power consumption.
> > >
> > > ---
> > > [1] Yao M, Zhao G, Zhang H, et al. Attention spiking neural networks[J]. IEEE transactions on pattern analysis and machine intelligence, 2023, 45(8): 9393-9410.
> > >
> > > [2] Luo X, Yao M, Chou Y, et al. Integer-valued training and spike-driven inference spiking neural network for high-performance and energy-efficient object detection[C]//European Conference on Computer Vision. Cham: Springer Nature Switzerland, 2024: 253-272.
> > >
> > > [3] Yao M, Hu J K, Hu T, et al. Spike-driven Transformer V2: Meta Spiking Neural Network Architecture Inspiring the Design of Next-generation Neuromorphic Chips[C]//The Twelfth International Conference on Learning Representations.
> > >
> > > [4] Yao M, Richter O, Zhao G, et al. Spike-based dynamic computing with asynchronous sensing-computing neuromorphic chip[J]. Nature Communications, 2024, 15(1): 4464.
> > >
> > > [5] Yao M, Qiu X, Hu T, et al. Scaling spike-driven transformer with efficient spike firing approximation training[J]. IEEE Transactions on Pattern Analysis and Machine Intelligence, 2025.

---

> > > > ### Comment · Reviewer_dnLE · 2025-08-06
> > > >
> > > > I appreciate the authors further engagement. I am inclined to increase my score, though the camera ready revision should revise down the strength of claims, given no statistical significance tests or real world measurements are performed. I think its fair to say the proposed method offers comparable performance to existing methods, and opens an exciting direction to yield more substantial savings in the future.
> > > >
> > > > On the topic of power, my understanding of the [4] reference shared is the theoretical metric was used to understand what major factors should be considered for network design (such as how many layers), but real world testing was also performed. I think the disconnect is whether using power measurements for add/multiply operations from a specific technology (45nm, which is a decade old) counts as a theoretical & hardware-agnostic measurement. I would argue not. As semiconductor technology improves, we can conceive of a world where the ratio of cost between a multiply and an add decreases (so instead of a 5x difference, maybe only 2x). Therefore, reporting the raw estimated multiply and accumulate FLOPS (separately) would be the most theoretical & hardware-agnostic metric since its not biased towards a specific energy cost ratio.

---

> > > > > ### Author Response · Authors · 2025-08-06
> > > > >
> > > > > Thank you for the feedback, recognition, and for updating the score. We greatly appreciate your patience and the multiple rounds of communication, as your engagement has significantly enhanced the quality of our work. In the revised version, we will supplement specific numerical comparisons of mac and ac for reference, and indeed, the comparison of power consumption requires further development within the community.

---

### Official Review · Reviewer_gksX · 2025-07-03

**Clarity:** 3
**Significance:** 3
**Originality:** 3
**Rating:** 4
**Confidence:** 4

**Summary:**

This paper proposes SpikeFET, a fully spiking neural network (SNN) framework for unified frame-event object tracking. Built upon a dual-single-dual architecture, the framework effectively integrates convolutional and Transformer-based SNN modules for local and global modeling, while preserving the low-power advantages of SNNs. The introduced Randomized Patchwork Module (RPM) and Spatio-Temporal Regularization (STR) address key architectural limitations—namely translation invariance degradation due to convolutional padding and temporal misalignment between features. Experiments on three major benchmarks (FE108, VisEvent, COESOT) demonstrate that SpikeFET achieves state-of-the-art tracking performance with significantly lower power consumption.

**Questions:**

1. While the methodology is technically solid, the model relies heavily on pretraining with frame-based datasets (e.g., COCO, LaSOT), which raises concerns about its ability to generalize to event-only scenarios. If image frames are unavailable and only event data is used, can the model maintain its performance?  Can it generalize well?
2. Also, would the pre-trained model show different effects when using both datasets?

**Ethical Concerns:**

["NO or VERY MINOR ethics concerns only"]

**Final Justification:**

I appreciate the authors’ efforts in responding to the comments. I will maintain my first score.

**Limitations:**

yes

**Quality:**

3

**Strengths And Weaknesses:**

Strengths:

1. The first fully SNN-based framework for frame-event fusion tracking, with novelty and relevance.
2. The architecture is well-designed, combining convolutional and Transformer-style SNN components to exploit both spatial and temporal information.
3. RPM and STR modules are effective in addressing structural weaknesses and enhancing feature consistency.
4. The experimental section is thorough, with comparisons to strong baselines and detailed ablation studies.

Weaknesses:

1. While the methodology is technically solid, the model relies heavily on pretraining with frame-based datasets (e.g., COCO, LaSOT), which raises concerns about its ability to generalize to event-only scenarios.
2. If image frames are unavailable and only event data is used, it is uncertain whether the model can maintain its performance, and it is also unclear whether it can generalize well.
3. Additionally, it is not known whether the pre-trained model would show different effects when using both datasets.

---

> ### Author Rebuttal · Authors · 2025-07-31
>
> We thank Reviewer gksX for the valuable comments.
> ## W1 and Q1: While the methodology is technically solid, the model relies heavily on pretraining with frame-based datasets (e.g., COCO, LaSOT), which raises concerns about its ability to generalize to event-only scenarios.
> We sincerely appreciate the reviewer’s insightful comments and apologize for any confusion caused by our previous unclear descriptions. We would like to clarify that our model does not rely solely on pre-training with frame-based datasets to achieve strong performance. In particular, for both SpikeFET-Tiny and SpikeET, we do not use frame-based pre-training, yet they outperform existing non-pre-training methods. For a fair comparison with approaches that do incorporate pre-training, such as ViPT [25] and SDSTrack [19], we pre-trained SpikeFET-Base using the same datasets—GOT-10k [43], TrackingNet [42], COCO [40], and LaSOT [41]. As a result, our model also achieved state-of-the-art performance on the COESOT [7] and FE108 [8] datasets.
>
> Additionally, there is no need to concern yourself about our model's generalization capabilities in purely event-based scenarios. As shown in Table 2, SpikeET—the event-based variant of SpikeFET—demonstrates superior performance on event-exclusive datasets even without pre-training, achieving state-of-the-art (SOTA) results.
> ## W2 and Q1: If image frames are unavailable and only event data is used, it is uncertain whether the model can maintain its performance, and it is also unclear whether it can generalize well.
> As demonstrated in W1 and Table 2, our model achieves superior performance on event-only data while exhibiting enhanced generalization capability. The improved generalization ability of our model on event stream data stems from the inherent suitability of SNNs for processing sparse event stream data, which enables more effective learning from event-based representations.
> ## W3 and Q2: Additionally, it is not known whether the pre-trained model would show different effects when using both datasets.
> Thank you for the insightful comment. We believe that, in theory, pre-training with frame-event paired datasets should yield better results than current methods. However, the open-source community currently lacks a substantial volume of such paired datasets, preventing us from empirically validating this hypothesis. Nevertheless, in our subsequent work, we will verify this using a large-scale dataset we are currently collecting and will release it, filling this gap in the open-source community.

---

> > ### Comment · Reviewer_gksX · 2025-08-03
> >
> > While some of my concerns have been addressed, I am unable to raise my score in light of the feedback provided by the other reviewers. I appreciate the authors’ efforts in responding to the comments.

---

> > > ### Author Response · Authors · 2025-08-03
> > >
> > > Dear Reviewer gksX,
> > >
> > > Thank you for your feedback. We are pleased to hear that our responses have addressed most of your concerns. We believe the additional experiments, analysis, and explanation have significantly improved the quality and clarity of our submission. We hope that you and other reviewers may regard this as a sufficient reason to raise the score.
> > >
> > > Best, Authors

---

### Official Review · Reviewer_kyP6 · 2025-07-05

**Clarity:** 3
**Significance:** 3
**Originality:** 3
**Rating:** 4
**Confidence:** 3

**Summary:**

This paper proposes SpikeFET, a fully spiking neural network (SNN)-based framework for frame-event object tracking. By addressing the limitations of high computational overhead and inefficient sparse event modeling, the authors introduce a Random Patchwork Module (RPM) to mitigate translation invariance degradation and a Spatial-Temporal Regularization (STR) strategy to improve similarity metrics. Experimental results demonstrate superior performance on multiple benchmarks (FE108, VisEvent, COESOT) while achieving significant power reduction.

**Questions:**

1. ​​RPM Implementation Details​​: How is the concatenation direction (horizontal/vertical) determined for RPM? Is this choice data-dependent or fixed?
​​2. Hyperparameter Sensitivity​​: Were α and β in the loss functions tuned on the validation set, or derived from prior work?

**Ethical Concerns:**

["NO or VERY MINOR ethics concerns only"]

**Final Justification:**

Because of the rebuttal, the score changes to BA

**Limitations:**

Overfitting to datasets with abundant event data may limit applicability to low-event scenarios.

**Quality:**

3

**Strengths And Weaknesses:**

Strengths​​:
1. RPM effectively reduces positional bias via randomized spatial reorganization and learnable type encoding.
2. STR enforces temporal consistency between template features, improving similarity metrics.
3. Strong performance gains on challenging datasets and comprehensive ablation studies.


​​Weaknesses​​:

1. The worse performance on VisEvent compared to ViPT and SDSTracker requires further analysis and raises concerns about dataset-specific optimizations.
2. Complex loss function design, which requires hyperparameter analysis.

---

> ### Author Rebuttal · Authors · 2025-07-31
>
> We thank reviewer kyP6 for the constructive comments, We provide our feedbacks as follows.
> ## W1: The worse performance on VisEvent compared to ViPT and SDSTracker requires further analysis and raises concerns about dataset-specific optimizations.
> Thank you for the comment. In reality, Spiking Neural Networks (SNNs) have lagged behind Artificial Neural Networks (ANNs) on most benchmarks in recent years, mainly because of their inherently sparse computations. However, as SNNs continue to evolve, the performance gap is steadily narrowing. With our proposed design, SNNs have achieved comparable or even superior performance to ANNs on object tracking tasks.
>
> Additionally, we posit that SpikeFET is particularly well-suited for processing large-scale datasets and data containing challenging scenarios. In these challenging conditions, the inherent advantages of event cameras become evident, allowing SpikeFET to demonstrate superior performance. In contrast, ViPT and SDSTrack excel in less demanding scenarios. As shown in Table 1, it can be observed that SpikeFET-Base achieved increases of 0.2%, 2.2%, and 4.5% in AUC metrics compared to SpikeFET-Tiny on the FE108 [8], VisEvent [6], and COESOT [7] datasets (arranged in ascending order of data volume), with the performance gap progressively widening. Table 2 reveals that on FE108 [8]—a dataset featuring numerous challenging scenarios (e.g., overexposure, low illumination)—SpikeET's performance is fully leveraged: its AUC is only 4.1% lower than SpikeFET-Tiny, while outperforming all event-based tracking algorithms. In contrast, it underperforms by 17.4% on VisEvent.
>
> Based on this, SpikeFET outperforms ViPT [25] and SDSTrack [19] on COESOT [7] and FE108[8], while slightly underperforming on VisEvent[6] with only a marginal performance drop. Additionally, it exhibits lower power consumption compared to ViPT [25] and SDSTrack [19], which stands as one of SpikeFET's key advantages. Corresponding analyses can be found in lines 251-256. Due to space constraints in the main text, we did not elaborate further, but we will add this in the revised version.
> ## W2: Complex loss function design, which requires hyperparameter analysis.
> Due to the loss function paradigm for object tracking [3], we accordingly selected six loss functions from two modalities, as shown in Eq. (13), and retained the same hyperparameters. In addition, we introduced two newly proposed loss functions, $L_{res}$ and $L_{sim}$, resulting in a total of eight loss functions, as detailed in Eq. (14). The corresponding hyperparameter analysis can be found in Appendix E, where we present ablation studies only in the vicinity of the optimal parameters. The ablation experiments validate our hypotheses. This will be clarified in the final version.
> ## Q1: RPM Implementation Details​​: How is the concatenation direction (horizontal/vertical) determined for RPM? Is this choice data-dependent or fixed?​​
> (1) The RPM we proposed determines the connectivity direction for each batch through random selection, where every batch may choose either horizontal or vertical orientation. We must align every sample within the same batch to the same direction so that they can be properly stacked into that batch. Such randomization can create greater positional variability.
> (2) The selection of connection directions is randomly chosen for any data—it is neither data-dependent nor predetermined.
> ## Q2: Hyperparameter Sensitivity​​: Were α and β in the loss functions tuned on the validation set, or derived from prior work?
> The $\alpha$ and $\beta$ hyperparameters in our loss function were both tuned on the validation set, primarily for adjusting model stability. To maintain consistency with the magnitude range of hyperparameters in object tracking, we performed tuning within the range of 0.1 to 5.
>
> When $\alpha$ is excessively large, the fusion results become overly dependent on the poorer-performing modality. When $\alpha$ is too small, the model cannot achieve sufficient fusion. When $\beta$ is excessively large, it overly forces inconsistent features to align. When $\beta$ is too low, it fails to fully achieve the STR effect. After tuning, we selected $\alpha=1$ and $\beta=0.5$ as the final parameters for the model.
> ## Limitations: Overfitting to datasets with abundant event data may limit applicability to low-event scenarios.
> Thank you very much for your question. We believe that in low-event scenarios, relying solely on the event modality would lead to poor tracking performance. Given this limitation, we introduced frames to compensate for the missing texture information in event data. Our proposed SpikeFET architecture effectively fuses image and event features within an SNN, fully leveraging the advantages of both modalities. As a result, SpikeFET achieves robust performance not only in low-event scenarios such as static or slow-moving scenes, but also in high-event scenarios like fast motion, thereby overcoming the performance degradation observed when using event data alone. We will add these visualization results in the revised version.

---

> > ### Comment · Reviewer_kyP6 · 2025-08-07
> >
> > Thanks for the detailed rebuttal, it partially addresses my concerns.

---

> > > ### Author Response · Authors · 2025-08-07
> > >
> > > We are delighted that our rebuttal has addressed your concerns. We sincerely appreciate your recognition of our work.

---

### Comment · Area_Chair_tfD7 · 2025-08-06
**Please acknowledge and respond to the author rebuttal as soon as possible**

Dear reviewers,

A last and urgent reminder for those who have not yet done so: please acknowledge and respond to the author rebuttal as soon as possible, so that they still might have time to respond.

Kind regards,

AC.

---

### Decision · Program_Chairs · 2025-09-17

**Decision:**

Accept (poster)

**Comment:**

The authors introduce a fully spiking tracking method based on frames and events, SpikeFET. They introduce two mechanisms, Randomized Patchwork Module (RPM) and Spatio-Temporal Regularization (STR), to improve translation invariance and better temporal feature alignment. The results are state of the art.

The reviewers raised several issues, from unclarity on the fairness of comparisons to whether and how pretraining is performed based on frames, the choosing of the template frames, and whether improvements come from more data only or the RPM. These issues were successfully addressed by the authors.

A discussion evolved on the measurement of the power savings by using an SNN for the frame-event fusion. The authors base themselves on well-established theoretical measures, counting MACs and ACs. A reviewer remarks that the numbers are based on old technology. Moreover, it has been remarked that an implementation on neuromorphic hardware and actual measurements would be better. This is true, but the procedure does align with common practice in the field.

Another issue is the explanation of the large difference between the performance with and without RPM, more than doubling the AUC and PR. The authors put this on the detrimental influence of the padding on translation invariance. Although this may be true, more in-depth analysis of this matter would be desirable. This could involve other data sets, or padding not with the searched for pattern but with other, unrelated data.

All in all, the reviewers find most of their main concerns addressed.